# VIDEO2REACTION: MAPPING VIDEO TO AUDIENCE REACTION DISTRIBUTION IN THE WILD

## ABSTRACT

Understanding audience reactions to video content is crucial for improving content creation, recommendation systems, and media analysis. We introduce **Video2Reaction**, a multimodal dataset that maps short movie segments to the *distributional induced emotional reactions* of viewers in the wild, as expressed through social media. Unlike most prior datasets that focus on *perceived emotions* (i.e., the emotions portrayed by characters in a movie clip), **Video2Reaction** centers on the induced emotions triggered by the movie clip. Additionally, we model these reactions as *distributions* over categorical emotions, rather than reducing them to a single dominant label, enabling fine-grained learning of collective emotional responses. **Video2Reaction** can support a range of applications, including audience reaction prediction for new video content, emotion-aware video retrieval, and content optimization based on expected viewer engagement. By providing a comprehensive benchmark for distributional video-to-reaction modeling, **Video2Reaction** advances the study of audience engagement and emotional impact in multimedia content. The dataset is available at `https://huggingface.co/datasets/video2reac/Video2Reaction`.

## 1 INTRODUCTION

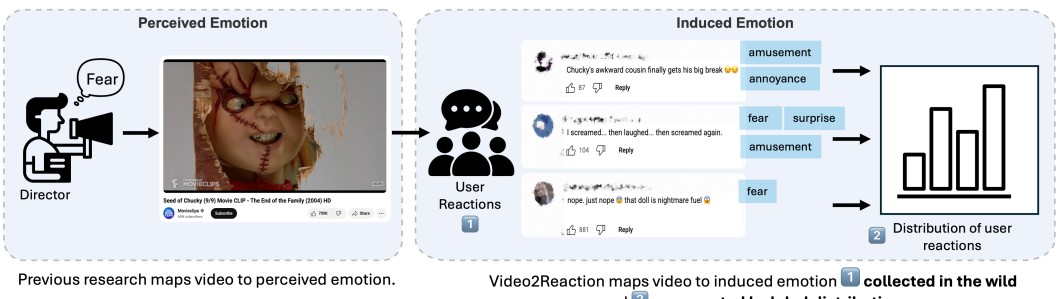

Figure 1: **Video2Reaction** is the first benchmark that uses video data to directly learn induced emotion distribution in the wild.

Understanding how people react to video content is a crucial yet underexplored aspect of affective computing. Prior work has established a distinction between perceived emotion—the emotion conveyed or expressed by the content itself—and induced emotion—what is experienced by the audience (Tian et al., 2017). While existing video sentiment datasets focus on perceived emotions (e.g., emotions of characters in the scene or filmmaker's emotion intent), audience reactions can vary greatly depending on personal, cultural, or temporal context. For example, Figure 1 illustrates how a horror movie scene might be perceived as frightening or suspenseful based on the director's intent, but viewer reactions could diverge significantly-some may experience genuine fear and anxiety, others might laugh at predictable genre tropes, and those with personal trauma related to similar situations might be triggered and distressed. This underscores the need to study induced emotional reactions directly, rather than relying solely on perceived sentiment.

In this paper, we introduce **Video2Reaction**, a multimodal dataset consisting of over 10,000 movie clips, paired with audience reaction distribution derived from viewer comments, allowing for a precise mapping between visual content and the emotional reactions it induces. To our knowledge, **Video2Reaction** is the largest multimodal dataset capturing induced emotion, and is the first dataset capturing distribution of induced emotion from cinematic content in non-controlled environments (in the wild). Unlike perceived emotions, which are typically modeled as unimodal (single-label), induced emotions can be either unimodal or distributed across multiple emotions. This distinction makes it more important to learn the distribution of reactions, rather than simply predicting single or multi-class labels. To address this, we frame audience emotion recognition as a **label distribution learning** (LDL) problem (Geng, 2016). Rather than classifying a single dominant reaction, we model the distribution of emotional reactions across viewers for each video, enabling us to capture the diverse and nuanced nature of audience reactions.

Our main contributions are as follows:

- We introduce **Video2Reaction**, a high-quality dataset that maps movie scenes to distributions over audience's induced emotions in the wild, grounded in real-world viewer comments. To our knowledge, **Video2Reaction** is the largest multimodal dataset capturing induced emotion (10,348 videos spanning approximately 400 hours and 800,000 comments), and the first dataset capturing induced emotion from cinematic content in the wild.

- We develop a scalable two-stage multi-agent LLM-based annotation pipeline that enables cost-effective, extensible reaction labeling, paving the way for future dataset updates as new content and evolving audience perspectives emerge.

- We propose a novel and challenging benchmark: predicting audience reaction distributions from multimodal video content, and design a comprehensive evaluation framework that captures both distributional alignment and dominant reaction classification.

- We benchmark a diverse range of approaches, including classical LDL algorithms, adapted multimodal emotion recognition models, and foundation generative models, highlighting their strengths and limitations in forecasting human reactions towards movie content.

## 2 RELATED WORK

### 2.1 EMOTION VIDEO DATASETS

Table 1: Comparison of **Video2Reaction** and Existing Emotion Prediction Video Datasets

| Dataset | # Videos (Hours) | Emotion Type | Emotion Annotation Setting | Emotion Representation |
|---|---|---|---|---|
| IEMOCAP (Busso et al., 2008) | 7,433 (12 hours) | Perceived | Lab | Single-label |
| MELD (Poria et al., 2018) | 13,000 (13 hours) | Perceived | Lab | Single-label |
| CMU-MOSEI (Zadeh et al., 2018) | 23,453 (66 hours) | Perceived | Lab | Continuous, Single-label |
| LIRIS-ACCEDE (Baveye et al., 2015) (Muszynski et al., 2019) | 9, 800 (27 hours) | Perceived, Induced | Lab | Continuous |
| COGNIMUSE (Zlatintsi et al., 2017) | 50 (3.5 hours) | Induced | Lab | Continuous, Single-label |
| DEAP (Koelstra et al., 2011) | 120 (2 hours) | Induced | Lab | Continuous |
| CMSV (Xu et al., 2024) | 8,210 (69 hours) | Induced | In the wild | Single-label |
| **Video2Reaction** (ours) | 10,348 (398 hours) | Induced Emotion | In the wild | Label Distribution |

Table 1 summarizes key characteristics of existing emotion video datasets and **Video2Reaction**. Most prior datasets like IEMOCAP (Busso et al., 2008), MELD (Poria et al., 2018), and CMU-MOSEI (Zadeh et al., 2018) focus on *perceived emotions*—the emotions expressed or felt by the characters in a movie scene—rather than the *induced emotions* experienced by the audience.

Some datasets like LIRIS-ACCEDE (Baveye et al., 2015), COGNIMUSE (Zlatintsi et al., 2017), and DEAP (Koelstra et al., 2011) have attempted to capture induced emotion. However, those datasets only capture emotional response in the 2D valence-arousal space, whereas our dataset captures the full distribution of categorical emotions. Furthermore, prior datasets often rely on controlled environments, such as lab settings or small groups of participants watching content together (Muszynski et al., 2019; Tian et al., 2017), limiting the impact across larger and more diverse demographics. For instance, Muszynski et al. (2019) extended LIBRIS-ACCEDE database (Baveye et al., 2015) to use aesthetic features from movie scenes to model induced emotions, but only with 10 participants in a co-viewing setting—unlike the solo, asynchronous media consumption typical today. Xu et al.

(2024) is a recent benchmark that attempts to capture induced emotion in the wild but their task is to predict induced emotion given a pair of video and comments while our benchmark predict induced emotion from video content only.

Additionally, existing emotion video datasets typically aggregate reactions into a single outcome label (e.g., a majority vote or mean score) for each clip (Koelstra et al., 2011; Baveye et al., 2015; Zlatintsi et al., 2017), failing to reflect the diversity of viewer responses. In contrast, we model the full distribution of emotional reactions, using soft labels derived from real-world audience responses.

## 2.2 LABEL DISTRIBUTION LEARNING

Label Distribution Learning (LDL) was formalized by Geng (2016) as a framework to address label ambiguity problems in tasks such as age estimation and sentiment prediction. Unlike traditional classification, which assumes a single ground truth label per instance, LDL represents each instance with a distribution over labels, capturing uncertainty and correlation among neighboring labels.

Existing emotion-related LDL datasets included LDL variants of Twitter and Flickr datasets (Yang et al., 2017) and Emotion6 (Peng et al., 2015), in which ground truth label distribution is labeled by a limited number of annotators. However, these datasets do not capture video data.

LDL methods are generally categorized into three families (Geng, 2016): problem transformation (PT), algorithm adaptation (AA), and specialized algorithms (SA). PT methods reformulate LDL as a single-label learning task by resampling training set into single-labeled instances using the target multilabel distribution. AA approaches modify existing learning algorithms to accommodate the label distribution format. For instance, AA-kNN (Geng, 2016) extends k-nearest neighbors by using neighbor distances to estimate label distributions. This category provides a flexible way to adapt traditional single-label or multi-label classification techniques to LDL scenarios. SA include algorithms explicitly designed for LDL. These methods often incorporate custom mechanisms to exploit unique aspects of label distributions. For example, SA-BFGS (Geng, 2016) uses an effective quasi-Newton method BFGS to optimize KL divergence objective. LDL-LRR (Jia et al., 2019) introduces a label ranking-aware loss function to better capture ordinal relationships.

The benchmark we adopt to validate our dataset builds on existing LDL methods, extending them to incorporate foundation Vision-Large Language Models as an additional class of algorithms capable of learning distribution alignment (Meister et al., 2024).

## 3 VIDEO2REACTION DATASET

The **Video2Reaction** dataset has been developed to facilitate the prediction of audience reactions from short movie segments. Each instance consists of a movie clip paired with a probability distribution over possible audience reactions (listed in Table 2). To facilitate future research on raw video analysis, we provide YouTube video IDs for all clips. Additionally, we release preprocessed features, including state-of-the-art visual embeddings and audio embeddings extracted using pretrained models. Details on video data preprocessing and the annotation of audience reaction distributions are provided in Section 3.2 and Section 3.3.

Table 2: Finer-grained reaction categories (21) used in **Video2Reaction** grouped by sentiment.

| Sentiment Category | Finer-grained Reaction Categories |
|---|---|
| Positive | amusement, excitement, joy, caring, admiration, relief, approval |
| Negative | fear, nervousness, embarrassment, disappointment, sadness, grief, disgust, anger, annoyance, disapproval |
| Ambiguous | realization, surprise, curiosity, confusion |

## 3.1 DATA COLLECTION

We curate movie clips from the CondensedMovies dataset (Bain et al., 2020) (licensed with CC BY 4.0), which contains licensed content from the *Movieclips*[1] YouTube channel. The use of licensed

---

[1]https://www.youtube.com/@MOVIECLIPS

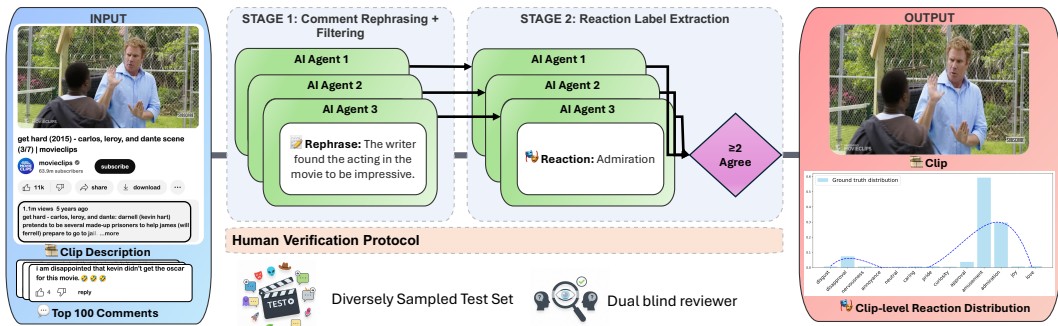

Figure 2: **Overview of Video2Reaction Two-Stage LLM-based Data Annotation Pipeline.** *Stage 1* rephrases comments to explicitly state their reactions towards the clip. It also filters out comments that lack a discernible reaction to the clip. *Stage 2* extracts reaction labels, with majority voting across three LLM agents to ensure consistency and discard ambiguous cases.

content improves the longevity of the dataset, as these clips are less likely to be removed from the platform. To ensure meaningful audience engagement, we retain only videos with a minimum of 10,000 views and at least 10 comments. The selected clips, originally uploaded between 2011 and 2019, are downloaded for further processing. Viewer comments on these videos extend through 2025, resulting in each clip having a minimum of six years of audience commentary. This broad temporal range enables the dataset to capture evolving audience perspectives and contemporary references, supporting a more comprehensive and generalizable benchmark for reaction prediction.

## 3.2 VIDEO DATA PREPROCESSING

Each movie clip in **Video2Reaction** is segmented into key scenes using PySceneDetect's content adaptive detection algorithm [2]. For each scene, we extract the following features:

**Visual Features**: ViT embeddings (Wu et al., 2020) of the middle frame of each scene.

**Audio Acoustic Features**: CLAP embeddings (Wu et al., 2022), pre-trained on a mixture of sounds.

**Audio Semantic Features**: HuBERT embeddings (Hsu et al., 2021), pretrained on speech only data.

In addition to temporal audio-visual features, we also include **Clip Description**, a short clip description provided by the Youtube channel, preprocessed using BERT-based text embeddings (Devlin et al., 2018). Detailed description of additional dataset metadata is available in Appendix A.1.

## 3.3 REACTION DISTRIBUTION ANNOTATION

**Reaction Taxonomy.** To represent the complexity of audience reaction, we initially adopt the 28-category emotion taxonomy from GoEmotions (Demszky et al., 2020), which was originally designed for Reddit comments and aligns naturally with our YouTube-based dataset. As a social media platform similar to Reddit, YouTube also features a wide range of audience interactions and emotional responses, making the GoEmotions taxonomy a natural fit. However, we drop 7 of the original categories in GoEmotions due to their significant under-representation in our data. These 7 reactions contribute to less than 0.01% of the distribution mass on average. Our final taxonomy consists of 21 fine-grained emotions (Table 2). The definition for each category is in Appendix A.2.

**Two-stage Multi-agent Reaction Annotation Pipeline.** Figure 2 outlines our LLM-based pipeline for annotating audience reactions based on user comments. Each comment is processed in two stages. The first stage of rephrasing comments is critical, as many audience comments reflect implicit reactions or off-topic remarks that need contextual interpretation to reveal their emotional intent. For example, a comment like *"I'm so disappointed this actor didn't win an Oscar"* will be mistakenly labeled as *disappointment* without being rephrased as *"the acting in the movie is so impressive"* in the first stage. Prompt details for both stages are provided in Appendix B.1.

---

[2]https://www.scenedetect.com/

Given the volume of raw comments and the non-stationary nature of induced emotions over time, we designed a scalable and reliable multi-agent annotation pipeline that supports frequent dataset updates, ensuring long-term relevance and impact. Motivated by prior findings in the use of multi-agent framework in text classification (Trad & Chehab, 2024), compound systems (Chen et al., 2024), and chain-of-thought reasoning (Wang et al., 2023; Choi et al., 2024), we employ an ensemble of of three medium-sized multilingual instruction-tuned LLMs [3] and adopt a straightforward majority voting approach for our emotion annotation task.

**Quality of the reaction annotation.** To account for annotation noise from human raters when faced with a large number of fine-grained categories (as demonstrated by low interater correlation in past research Demszky et al. (2020)), we adopt a dual blind human verification protocol. First of all, we randomly sample 100 movie clips with balanced representation across all movie genres. From each clip, 10 comments are randomly selected, yielding a total of 1,000 comments for human evaluation. Then each comment is independently reviewed by two annotators. In cases of disagreement, a third annotator is consulted, and the final label is determined by majority vote. Overall, $86\%$ of the LLM-assigned reaction labels were judged to be correct, $7.8\%$ incorrect, and $6.2\%$ indeterminate (Table 9 in Appendix B.2).

Beyond overall accuracy, we analyze errors in the LLM-based annotation pipeline for future users' consideration and for future improvements. As shown in Table 10 (Appendix B.2), performance remains above 70% across all genres. However, genres such as Comedy and Drama pose greater challenges due to their reliance on subtle cues (e.g., pop culture references), which can obscure emotional tone even for human annotators.

## 3.4 DATASET STATISTICS

Table 3 summarizes Video2Reaction at the movie, clip, and comment levels. The dataset spans 390 hours of video from 1,545 movies and includes 10,348 clips. On average, each clip is segmented into 44 scenes and is associated with 24 viewer comments used to construct the reaction distribution—substantially more than prior LDL benchmarks such as *Twitter_LDL* and *Flickr_LDL* (Yang et al., 2017), which constructs a label distribution from 11 annotators only.

Importantly, Video2Reaction captures substantial variation in audience reactions within the same movie, with a median Chebyshev distance of $0.48$ (out of $1.0$) between clip-level distributions of the same movie. This highlights the need for clip-level rather than movie-level modeling.

As shown in Figure 3a, the dataset is highly imbalanced across the 21 reaction categories (with an imbalance factor of 28.36), a challenge addressed in our evaluation design (Sections 4.2 and 5). Figure 3b further shows that top-1 reaction probability varies considerably across clips (with a median of approximately $0.4$), underscoring the importance of modeling full reaction distributions instead of predicting only the dominant label.

Table 3: Descriptive statistics for video and comment data in the Video2Reaction dataset.

| Category | Total | Min | Mean | Median | Max |
|---|---|---|---|---|---|
| **Movie-level Statistics** | | | | | |
| Inter-segment Chebyshev distance in reaction distribution (out of 1.0) | 1,545 | 0.05 | 0.48 | 0.48 | 0.91 |
| **Clip-level Statistics** | | | | | |
| Clip Duration (sec) | 389.81 hrs | 23.15 | 135.61 | 127.60 | 367.36 |
| Key Scenes | 455,226 | 16 | 43.99 | 39.00 | 176 |
| **Comment-level Statistics** | | | | | |
| Raw Comments | 771,684 | 19 | 74.57 | 77.00 | 100 |
| Retained Comments | 252,462 | 10 | 24.40 | 22.00 | 100 |

---

[3] LLaMA-3.1-8B-Instruct (Grattafiori et al., 2024) , Qwen2.5-14B-Instruct (Team, 2024) , and DeepSeek-R1-Distill-Qwen-7B (DeepSeek-AI, 2025)

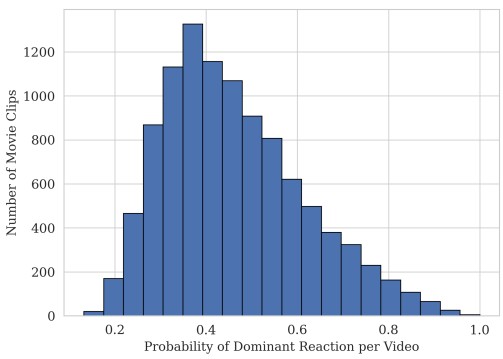

(a) Total number of videos and mean video-level probability of each reaction category ($\gamma$ denotes imbalance factor)

(b) Distribution of dominant reaction probability per video

Figure 3: Key Statistics on Reaction Outcome in the Video2Reaction dataset

## 4 BENCHMARK

### 4.1 PROBLEM FORMULATION

In this work, we frame the problem of audience reaction prediction as a label distribution learning (LDL) task. Unlike traditional classification, where the goal is to assign either a single label or a set of labels to each input instance, LDL seeks to predict a probability distribution over multiple labels, capturing the ambiguity and diversity in audience reactions.

Let $x$ denote an input video clip, which may contain visual, audio, and textual content. The audience reaction to $x$ is represented by a label distribution $\mathbf{d}_x = \{d_{xm}\}_{m=1}^{M}$, where $M$ is the number of affective reaction classes (e.g., amusement, confusion, fear, etc.), and each $d_{xm} \in [0, 1]$ indicates the proportion of annotators or viewers who associated label $m$ with the clip $x$. The label distribution satisfies the normalization constraint: $\sum_{m=1}^{M} d_{xm} = 1$. This distribution can be interpreted as the conditional probability $p(m|x)$ of observing reaction $m$ given video $x$. Our objective is to learn a model $f_\theta(x)$ that predicts a distribution $\hat{\mathbf{d}}_x$ approximating the empirical distribution $\mathbf{d}_x$.

Importantly, in our setting, $d_{xm}$ does not represent a soft target for a "correct" label in the traditional sense. Instead, it reflects the proportion of audience that has certain reaction given the video input.

### 4.2 EVALUATION METRICS

Our benchmark is constructed along two complementary axes: (1) **full distribution evaluation**, which assesses how well the predicted distribution over all possible reactions matches the groundtruth distribution; and (2) **dominant reaction evaluation**, which focuses on how accurately the model identifies and estimates the probabilities of the most dominant reactions. The second axis is particularly relevant for real-world applications—such as content recommendation, moderation, or trailer editing—which depend primarily on anticipating the strongest or most likely viewer responses rather than capturing the full reaction spectrum. For each axis, we carefully select a suite of metrics designed to capture different types of errors that are important for our task, allowing future research to prioritize specific evaluation criteria based on downstream use cases. Table 4 summarizes all metrics used in the benchmark, along with their formulas and the error types they capture.

**Full Distribution Evaluation.** Following (Geng, 2016), we evaluate how closely the predicted reaction distribution matches the ground-truth distribution using a suite of statistical distribution distance metrics. Since some metrics (i.e. Canberra and Clark) capture very similar types of errors for our task, we decided to include in the benchmark three distance-based metrics—Chebyshev, Clark, Kullback-Leibler (KL) and two similarity-based metrics—Cosine similarity and Intersection. In addition, inspired by (Wen et al., 2023), we incorporate an ordinal-aware evaluation by mapping the 21 reaction categories onto a valence-arousal-based emotional space. We then compute the Cumulative Absolute Distance (CAD) between the predicted and true *ordered* distributions. The metric assigns lower penalties to misclassifications involving emotionally similar reactions and higher penalties to

Table 4: Summary of evaluation metrics used in the Video2Reaction benchmark.

| Evaluation Category | Metric (Abbr.) | Formula | Error Type Captured |
|---|---|---|---|
| Full Distribution | Chebyshev (Che) ↓ | $\max_j \left| d_j - \hat{d}_j \right|$ | Largest per-class prediction error (worst-case mismatch) |
| | Clark (Cla) ↓ | $\sqrt{\sum_{j=1}^{c} \left( \frac{d_j - \hat{d}_j}{d_j + \hat{d}_j} \right)^2}$ | Errors on rare reactions (small denominators amplified) |
| | Kullback-Leibler (KL) ↓ | $\sum_{j=1}^{c} d_j \ln \frac{d_j}{\hat{d}_j}$ | Underestimation of true reactions, false-zero sensitivity |
| | Cumulative Absolute Distance (CAD) ↓ | $\sum_j \left| CDF(d_j) - CDF(\hat{d}_j) \right|$ | Distribution shifts ignoring ordinal relations |
| | Cosine (Cos) ↑ | $\frac{\sum_{j=1}^{c} d_j \hat{d}_j}{\sqrt{\sum_{j=1}^{c} d_j^2} \sqrt{\sum_{j=1}^{c} \hat{d}_j^2}}$ | Directional mismatch of prediction vs. groundtruth |
| | Intersection (Inter) ↑ | $\sum_j \min(d_j, \hat{d}_j)$ | Non-overlapping reaction prediction error |
| Dominant Reactions | Mean Reciprocal Rank (MRR) ↑ | $\frac{1}{r}$ | Incorrect ranking of dominant reaction ($r$ is the predicted rank of the target dominant reaction) |
| | Top-1 Probability Error (TPE) ↓ | $\left| \hat{d}_{j^*} - d_{j^*} \right|, \quad j^* = \arg\max_j d_j$ | Probability misestimation of dominant reaction |
| | Top-$k$ F1 (weighted) ($F1_k$) ↑ | $\sum_j \frac{n_j}{N} \cdot F1_{j,k}$ | Both precision and recall-based errors for top-$k$ reactions |

those involving emotionally distant categories, encouraging models to make semantically coherent predictions.

**Dominant Reaction Evaluation.** Unlike traditional emotion classification benchmarks, which often focus on unimodal distributions centered on a single perceived emotion, audience reactions in our task are multimodal, reflecting the diversity of viewer responses (as seen in Figure 3b). Therefore, we evaluate not only the top-1 prediction (single-label) but also the task's performance as a multilabel classification problem.

For single-label evaluation, we report the F1 score, Mean Reciprocal Rank (MRR), and Top-1 Probability Error (TPE). For multi-label evaluation, we compute F1 scores based on the top-k emotions from both the ground truth distribution and the model predictions. All F1 metrics are class-weighted to account for label imbalance.

## 4.3 COMPARATIVE MODELS

Building on the taxonomy of label distribution learning (LDL) algorithms proposed by (Geng, 2016), we organize comparative models into three categories—*Problem Transformation*, *Specialized Algorithms*, and *Algorithm Adaptation*—and extend the framework by introducing a fourth category to evaluate the zero-shot capabilities of modern foundation vision-language models.

**Problem Transformation (PT).** These methods reduce the LDL task to standard single-label learning (SLL) by resampling the training data. Each training instance $(\mathbf{x}_i, \mathbf{d}_i)$ with a label distribution $\mathbf{d}_i$ over $c$ classes is converted into multiple single-label examples $(\mathbf{x}_i, y_j)$ by sampling $y_j$ for class $j$ from $\mathbf{d}_i$. We include PT-Bayes (Geng, 2016), which uses a Bayes classifer and LDSVR (Geng & Hou, 2015), which adapts support vector regression for multi-output probability prediction as two representative baselines.

**Specialized Algorithms (SA).** These methods are explicitly designed for Label Distribution Learning (LDL), typically incorporating task-specific loss functions or optimization techniques. We evaluate three representative algorithms: SA-BFGS (Geng, 2016), which directly optimizes KL divergence using the BFGS algorithm; and LDL-LRR (Jia et al., 2019) and TLRLDL (Kou12 et al., 2024), both of which enhance the training objective by exploiting multi-label correlations.

**Algorithm Adaptation (AA).** These models were originally designed for related tasks, such as multimodal sentiment analysis, and are adapted here for label distribution prediction. We include CubeMLP (Sun et al., 2022), CTEN (Zhang et al., 2023), and MMIM (Han et al., 2021) as representative models. Unlike specialized LDL methods that only use late fusion of all input features, these models incorporate modules specifically designed to learn cross-modal features.

**Foundation Vision-Language Models (VLMs).** Beyond the standard LDL taxonomy (Geng, 2016), we introduce a fourth category to assess the zero-shot performance of large vision-language models. Although not trained for LDL, these models are increasingly used as proxies for human judgment in applications such as agent-based simulations (Park et al., 2023) and

behavioral research (Hwang et al., 2023; Jiang et al., 2022). We evaluate two leading open-source models—LLaVA-Next-Video-7B (Zhang et al., 2024) and Qwen2.5-VL (Team, 2025)—in a prompted classification setting over fixed reaction labels. Following findings from (Meister et al., 2024), we also apply temperature scaling using validation data to improve probability calibration.

**Implementation Details.** The PT, SA, and AA algorithms are trained on the dataset using 5 random seeds, and we report the mean and standard deviation of their performance in Tables 5 and 6. For foundation VLMs, the outputs are deterministic, as we directly use the model's predicted probabilities without additional training or stochastic components. Further details on algorithm-specific implementation are listed in Appendix C.

## 5 RESULTS

### 5.1 FULL DISTRIBUTION PREDICTION IS LEARNABLE BY BOTH SPECIALIZED AND ADAPTED ALGORITHMS

Table 5 shows that SA-BFGS (Geng, 2016), a classical specialized LDL algorithm, achieves the best performance across most distribution-based metrics, outperforming newer adaptation methods except for the Clark metric. However, the gap between SA-BFGS and the best adapted model (CTEN) is modest, indicating that adapted models can also effectively approximate audience reaction distributions.

Table 5: Full Distribution Evaluation Benchmark Results.

| Model Name | Cheb ↓ | KL ↓ | Cla ↓ | Cad ↓ | Inter ↑ | Cos ↑ |
|---|---|---|---|---|---|---|
| **Foundation VLM** | | | | | | |
| LLava-Next-Video-7B | 0.4110 | 1.5547 | 3.9710 | 4.3269 | 0.2970 | 0.4185 |
| Qwen2-VL | 0.3985 | 1.5216 | 3.9888 | 4.0333 | 0.3140 | 0.4401 |
| **Problem Transformation** | | | | | | |
| PT_Bayes | $0.9668 \pm 0.0219$ | $21.1994 \pm 0.0438$ | $2.7268 \pm 0.0477$ | $6.9506 \pm 0.2007$ | $0.0144 \pm 0.0009$ | $0.0272 \pm 0.0015$ |
| LDSVR | $0.2584 \pm 0.0000$ | $4.9794 \pm 0.0000$ | $\mathbf{2.1272} \pm 0.0000$ | $2.8912 \pm 0.0000$ | $0.6146 \pm 0.0000$ | $0.7872 \pm 0.0000$ |
| **Specialized Algorithms** | | | | | | |
| SA_BFGS | $\mathbf{0.2306} \pm 0.0011$ | $\mathbf{0.5976} \pm 0.0044$ | $3.9147 \pm 0.0012$ | $\mathbf{2.6711} \pm 0.0146$ | $\mathbf{0.6254} \pm 0.0015$ | $\mathbf{0.8089} \pm 0.0017$ |
| LDL_LRR | $0.3293 \pm 0.0028$ | $2.2569 \pm 0.0850$ | $4.1227 \pm 0.0007$ | $3.5177 \pm 0.0471$ | $0.5242 \pm 0.0035$ | $0.7115 \pm 0.0127$ |
| TLRLDL | $0.3368 \pm 0.0000$ | $7.9606 \pm 0.0014$ | $3.2362 \pm 0.0001$ | $3.8317 \pm 0.0001$ | $0.4264 \pm 0.0000$ | $0.5968 \pm 0.0000$ |
| **Algorithm Adaptation** | | | | | | |
| CubeMLP | $0.2738 \pm 0.0002$ | $0.6900 \pm 0.0003$ | $3.9669 \pm 0.0004$ | $3.2122 \pm 0.0003$ | $0.5624 \pm 0.0005$ | $0.7513 \pm 0.0000$ |
| CTEN | $0.2432 \pm 0.0017$ | $0.6033 \pm 0.0044$ | $3.9542 \pm 0.0033$ | $2.8277 \pm 0.0116$ | $0.6071 \pm 0.0021$ | $0.7977 \pm 0.0013$ |
| MMIM | $0.2442 \pm 0.0021$ | $0.6076 \pm 0.0046$ | $3.9593 \pm 0.0009$ | $2.8548 \pm 0.0349$ | $0.6019 \pm 0.0023$ | $0.7946 \pm 0.0027$ |

Moreover, the substantial performance gap between algorithms trained on the dataset and zero-shot approaches suggests that full distribution prediction from movie content is a learnable task, validating the feasibility of the benchmark. While specialized LDL algorithms currently lead, adapted models demonstrate strong potential and may close the gap with further task-specific tuning and optimization, offering more flexible and scalable solutions.

To provide more intuitive insights beyond aggregate metrics, Figure 4 in Appendix D.1 presents visual comparisons of predicted vs. ground-truth distributions for five random samples with varying entropy levels. Both SA-BFGS and CTEN are able to model groundtruth distribution shapes across different entropy levels but tend to underestimate the leading reaction's probability, particularly for more unimodal distributions. On average, the best model (SA-BFGS) exhibit a maximum per-class absolute error of 0.23 in estimating reaction probabilities, highlighting the challenge of fine-grained distribution modeling.

### 5.2 SPECIALIZED ALGORITHMS SHOW MODERATE PERFORMANCE AT DOMINANT REACTION CLASSIFICATION, BUT LONG-TAIL CHALLENGES REMAIN

There is a bigger gap between adapted algorithms and specialized algorithms in dominant reaction classification benchmark. As shown in Table 6, SA-BFGS achieves the best performance across most dominant reaction evaluation metrics, including weighted F1 Top 1, Mean Reciprocal Rank (MRR), and Top-1 Probability Error (TPE). However, while SA-BFGS is most effective at correctly

Table 6: Dominant Reaction Evaluation Benchmark Results.

| Model Name | TPE ↓ | MRR ↑ | F1 Top 1 (weighted) ↑ | F1 Top 2 (weighted) ↑ | F1 Top 3 (weighted) ↑ |
|---|---|---|---|---|---|
| **Foundation VLM** | | | | | |
| LLava-Next-Video-7B | 0.4103 | 0.1992 | 0.0143 | 0.1167 | 0.1374 |
| Qwen2-VL | 0.3923 | 0.3088 | 0.1958 | 0.2531 | 0.3037 |
| **Problem Transformation** | | | | | |
| PT-Bayes | $0.9661_{\pm 0.0000}$ | $0.1535_{\pm 0.0133}$ | $0.0001_{\pm 0.0000}$ | $0.0031_{\pm 0.0013}$ | $0.0741_{\pm 0.0676}$ |
| LDSVR | $\mathbf{0.1599}_{\pm 0.0000}$ | $0.7054_{\pm 0.0000}$ | $0.5034_{\pm 0.0000}$ | $0.5865_{\pm 0.0000}$ | $0.5696_{\pm 0.0000}$ |
| **Specialized Algorithms** | | | | | |
| SA-BFGS | $0.1882_{\pm 0.0021}$ | $\mathbf{0.7163}_{\pm 0.0038}$ | $\mathbf{0.5283}_{\pm 0.0053}$ | $\mathbf{0.6075}_{\pm 0.0014}$ | $\mathbf{0.6265}_{\pm 0.0026}$ |
| LDL-LRR | $0.2496_{\pm 0.0006}$ | $0.6700_{\pm 0.0130}$ | $0.4965_{\pm 0.0076}$ | $0.5684_{\pm 0.0038}$ | $0.5803_{\pm 0.0025}$ |
| TLRLDL | $0.2759_{\pm 0.0000}$ | $0.5559_{\pm 0.0002}$ | $0.4516_{\pm 0.0003}$ | $0.4895_{\pm 0.0001}$ | $0.4858_{\pm 0.0000}$ |
| **Algorithm Adaptation** | | | | | |
| CubeMLP | $0.2509_{\pm 0.0004}$ | $0.5996_{\pm 0.0000}$ | $0.2376_{\pm 0.0000}$ | $0.4399_{\pm 0.0000}$ | $0.5587_{\pm 0.0000}$ |
| CTEN | $0.2081_{\pm 0.0022}$ | $0.6939_{\pm 0.0022}$ | $0.4826_{\pm 0.0062}$ | $0.5950_{\pm 0.0022}$ | $0.5109_{\pm 0.0013}$ |
| MMIM | $0.2141_{\pm 0.0070}$ | $0.6749_{\pm 0.0058}$ | $0.4503_{\pm 0.0090}$ | $0.5728_{\pm 0.0037}$ | $0.5775_{\pm 0.0014}$ |

identifying the dominant reaction, LDSVR, a problem transformation algorithm, achieves the best probability estimation, with a lower TPE of 0.1599 compared to SA-BFGS's 0.1882.

Despite achieving a top-3 weighted F1 score of 0.62—moderate given the 21 reaction categories—current methods still fall short of ideal performance. This reflects the inherent difficulty of fine-grained emotion classification, where rare reactions are often underrepresented and harder to predict. A detailed class-wise performance breakdown is provided in the Appendix D.2.

## 5.3 FOUNDATION VLMS UNDERPERFORM IN ZERO-SHOT AUDIENCE REACTION PREDICTION

Table 5 and 6 shows that even with temperature scaling, foundation Vision-Language Models (VLMs) such as LLaVA-Next-Video-7B and Qwen2.5-VL significantly underperform compared to specialized LDL algorithms and adapted models. These models struggle with both identifying the correct dominant reactions and estimating their associated probabilities, highlighting their limitations in capturing the fine-grained, subjective nuances of audience responses. While some performance gap is expected in the zero-shot setting, the magnitude of this gap is larger than anticipated. This suggests that these large-scale models pretrained on a wide variety of human feedback data by default do not align well with human preference with respect to multimedia content.

## 6 CONCLUSION

In this work, we present **Video2Reaction**, the first benchmark for predicting fine-grained audience reaction distributions in the wild. Our benchmark is sustainably impactful, supported by a scalable annotation pipeline that enables iterative updates to adapt to the non-stationary nature of audience engagement.

We introduce a comprehensive evaluation framework with two complementary axes—full distribution prediction and dominant reaction estimation—and conduct extensive experiments across four categories of algorithms. Our findings demonstrate that this task is challenging yet feasible: specialized LDL algorithms currently achieve the best performance, but the gap between specialized and adapted models is notably smaller for full distribution prediction than for dominant reaction classification. Additionally, we observe a substantial performance gap for zero-shot foundation VLMs, highlighting the limitations of general-purpose models in forecasting multimedia audience reactions and the need for task-specific adaptation.

**Limitations.** While Video2Reaction offers a valuable resource for studying audience reactions and shows a great potential of forecasting audience reactions directly from media content, it has several limitations. First, the current reaction representations are derived solely from YouTube platform, which may not fully capture global audience behaviors. Future iterations will address this by expanding to multi-platform datasets, leveraging our scalable automatic annotation pipeline. Second, due to computational constraints, we have not yet explored fine-tuning foundation VLMs for this task, leaving their adaptation capabilities an open question.

**Ethics Statement.** We recognize that Video2Reaction may have both positive and negative societal implications. On the positive side, it can assist video creators in anticipating audience feedback prior to content release, potentially leading to improved content quality. However, in the absence of information about the demographic and psychographic characteristics of the audience from which the comments are collected, the predicted feedback may reflect biases, limiting its reliability and fairness. Additionally, to ensure users' privacy, we only release processed data and metadata. We do not release raw data that might include the information of Youtube commenters.

**Reproducibility Statement.** The full dataset with pre-defined train, validation, and test split are available at `https://huggingface.co/datasets/video2reac/Video2Reaction`. The github repository `https://github.com/wm-bit/video2reaction` includes code for video preprocessing, automated annotation pipeline and reproducing benchmark results for comparative algorithms. All the links are currently anonymized.

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

# A DATASET DETAILS

## A.1 DATASET METADATA

Table 7 summarizes key details of all features included in the dataset. Metadata features are recorded in the split-specific JSON files that can be downloaded on our Huggingface repository `https://huggingface.co/datasets/video2reac/Video2Reaction/tree/main`. Preprocessed Features and Reaction Outcome can be loaded directly from Huggingface or by using our custom Python script in `https://github.com/wm-bit/video2reaction`.

Table 7: Dataset Feature Details

| Feature | Description | Shape |
|---|---|---|
| **Metadata** | | |
| video_id | YouTube video identifier | - |
| imdbid | IMDb movie identifier | - |
| genre | List of movie genres | - |
| country | List of movie countries | - |
| movie_name | Name of the movie | - |
| clip_name | Name of the clip | - |
| clip_description | Description of the clip provided by the channel | - |
| **Preprocessed Features ($K$ denotes number of key frames)** | | |
| visual_feature | ViT embeddings[4] of the middle frame of each scene | (K, 768) |
| audio_acoustic_feature | CLAP embeddings[5], pre-trained on a mixture of sounds | (K, 1024) |
| audio_semantic_feature | HuBERT embeddings[6], pretrained on speech only data | (K, 1024) |
| clip_description_embedding | BERT-based text embeddings[7] for clip description | (768,) |
| movie_genre | One-hot encoding of movie genres | (23,) |
| **Reaction Outcome** | | |
| reaction_distribution | Distribution of viewer reactions (21 categories) | (21,) |

## A.2 REACTION CATEGORY DEFINITION

Table 8 provides definitions for each reaction category included in Video2Reaction dataset. The definitions are from GoEmotions taxonomy Demszky et al. (2020).

## A.3 DATA PREPROCESSING DETAILS

Each movie clip in Video2Reaction is segmented into key scenes using PySceneDetect's content adaptive detection algorithm[8]. The algorithm detects scene transition using rolling difference in HSL colorspace. We use a relatively low threshold 3.0 to segment the scenes so a fast-paced plot scene like a jump scare will be split into multiple detected scenes due to changes in the HSL colorspace. Each scene is represented using the middle frame.

# B DATA ANNOTATION PIPELINE

## B.1 IMPLEMENTATION DETAILS

Given the volume of raw comments to process, we employ an ensemble of three medium-sized multilingual instruction-tuned LLMs—LLaMA-3.1-8B-Instruct[9], Qwen2.5-14B-Instruct[10],

---

[8]`https://www.scenedetect.com/docs/latest/cli.html#detect-adaptive`
[9]`https://huggingface.co/meta-llama/Llama-3.1-8B-Instruct`
[10]`https://huggingface.co/Qwen/Qwen2.5-14B-Instruct`

Table 8: Video2Reaction Reaction Categories and Definitions. Definitions are copied from GoEmotions taxonomy Demszky et al. (2020)

| Sentiment | Reaction Category | Definition |
|---|---|---|
| Positive | Amusement | Finding something funny or being entertained. |
| | Excitement | Feeling of great enthusiasm and eagerness. |
| | Joy | A feeling of pleasure and happiness. |
| | Caring | Displaying kindness and concern for others. |
| | Admiration | Finding something impressive or worthy of respect. |
| | Relief | Reassurance and relaxation following release from anxiety or distress. |
| | Approval | Having or expressing a favorable opinion. |
| Negative | Fear | Being afraid or worried. |
| | Nervousness | Apprehension, worry, anxiety. |
| | Embarrassment | Self-consciousness, shame, or awkwardness. |
| | Disappointment | Sadness or displeasure caused by the nonfulfillment of one's hopes or expectations. |
| | Sadness | Emotional pain, sorrow. |
| | Grief | Intense sorrow, especially caused by someone's death. |
| | Disgust | Revulsion or strong disapproval aroused by something unpleasant or offensive. |
| | Anger | A strong feeling of displeasure or antagonism. |
| | Annoyance | Mild anger, irritation. |
| | Disapproval | Having or expressing an unfavorable opinion. |
| Ambiguous | Realization | Becoming aware of something. |
| | Surprise | Feeling astonished, startled by something unexpected. |
| | Curiosity | A strong desire to know or learn something. |
| | Confusion | Lack of understanding, uncertainty. |

and DeepSeek-R1-Distill-Qwen-7B[11]—chosen for their strong performance and favorable efficiency. All LLM agents share the same prompt for both stages, which are listed below.

The LLM annotation pipeline requires two following inputs:

- Clip Description, to set context to understand the sentiment of the comments. Our pipeline uses the short clip description provided by @MOVIECLIPS Youtube channel but we can also use a description generated by a video understanding model if no existing description is available.

- Comment, user-written comments on youtube.

---

[11]https://huggingface.co/deepseek-ai/DeepSeek-R1-Distill-Qwen-7B

864
865

**Stage 1: Rephrase and Filter Comment Prompt**

866
867
868
869
870
871
872

You are to roleplay as a senior director speaking to a junior director in training. You are reviewing comments from audience members on a variety of scenes from a variety of movies/shows. You are explaining to the junior director what the audience member is likely feeling due to the clip along with your reasoning. The goal is to teach the junior director how film can predictably be used to invoke certain emotions; as a result, you should ignore comments from audience members where the analysis shows there is likely nothing the junior director can generalize. Be concise in explanations, limit them to 2 sentences at most.

873
874
875
876

**Example 1:** `<description>` Paul makes a pair of thieves pay for bringing a knife to a gun fight. `<comment>` TTC Subways, this is Toronto these days. `<explanation>` The audience member here is comparing how Toronto subways seems similar to the subways in the scene. You cannot generalize the feelings of this audience member broadly so we will ignore this comment. `<rephrased>` None

877
878
879
880

**Example 2:** `<description>` Crocodile Dundee interrogates a gangster off the side of a building. `<comment>` It's Milton from Office Space. `<explanation>` The audience member just realizes the same actor from another TV show so there is no reaction towards any aspect of the clip here. `<rephrased>` None

881
882
883
884
885

**Example 3:** `<description>` Marius mourns his fallen comrades. `<comment>` My favorite song of this masterpiece `<explanation>` The audience member likes the music background in the movie clip so this is a reaction we want to know so that we can pay more attention to music and sound in the future. `<rephrased>` The writer loves the music background.

886
887
888

**Example 4:** `<description>` The Thénardiers swindle guests at their inn. `<comment>` The 1985 version is the best one yet `<explanation>` The audience member prefers another version of the movie but the comment does not explain why so we will ignore this comment. `<rephrased>` None

889
890
891
892
893
894

**Example 5:** `<description>` Walking alone at night, Paul comes face-to-face with an armed criminal. `<comment>` Keep shooting til he's dead, leave no "victim" to identify you later..and sue you.. `<explanation>` The audience member reiterates a character's action in the movie, implying that it is a good idea so this is an implicit reaction towards the character's decision or part of the plot in the scene. `<rephrased>` The writer agrees with the character's action in the scene.

895
896

Now, analyze the following comment: `<description>` {summary} `<comment>` {comment} Director:

897
898

899
900

**Stage 2: Extract Reaction Labels Prompt**

901
902
903
904

You are an assistant analyzing YouTube comments to extract audience reactions to a movie clip. Given the following inputs: - **clip_description**: A short description of the movie clip. - **rephrased_comment**: The original comment rewritten from a third-person perspective. Return a JSON object with the following fields:

905
906
907

- `high_level_reaction`: One or more words from {"joy", "sadness", "anger", "surprise", "disgust", "fear", "neutral"}.

908
909
910
911

- `finer_grained_reaction`: One or more words from {"admiration", "amusement", "anger", "annoyance", "approval", "caring", "confusion", "curiosity", "desire", "disappointment", "disapproval", "disgust", "embarrassment", "excitement", "fear", "gratitude", "grief", "joy", "love", "nervousness", "optimism", "pride", "realization", "relief", "remorse", "sadness", "surprise", "neutral"}.

912
913
914

- `reaction_reason_type`: One or more from {"cinematography", "character", "acting", "sound and music", "editing and pacing", "narrative and thematic elements", "personal"}; if no reason is clear, return `"none"`.

915
916
917

Return only a valid JSON object with these fields and **no additional text or explanations**.
**clip_description**: {clip_description}
**rephrased_comment**: {rephrased_comment}
**Output:**

## B.2 HUMAN EVALUATION & ADDITIONAL ERROR ANALYSIS

We provide a summary of human evaluation on our test set in Table 9. We further analyze different types of errors that our annotation pipeline tend to make and present them in Table 11 and 10.

Table 9: Human evaluation of automated reaction annotation on a sample of 1000 comments.

| Human Rating | Description | # comments (%) |
|---|---|---|
| Correct | Most annotators agree the LLM-assigned labels are correct. | 860 (86.0%) |
| Incorrect | Most annotators agree the LLM-assigned labels are incorrect. | 78 (7.8%) |
| Not Sure | Most annotators are unsure about the correctness of the labels (i.e. due to lack of context on movie references). | 62 (6.2%) |

Table 10: Performance across different movie genres.

| Movie Genre | % Correct | % Not Sure | % Incorrect |
|---|---|---|---|
| Adventure | 96.23 | 1.89 | 1.89 |
| Fantasy | 96.23 | 1.89 | 1.89 |
| Film-Noir | 95.00 | 5.00 | 0.00 |
| Musical | 93.94 | 0.00 | 6.06 |
| Documentary | 90.48 | 4.76 | 4.76 |
| History | 90.20 | 3.92 | 5.88 |
| Sci-Fi | 89.58 | 4.17 | 6.25 |
| Biography | 88.24 | 5.88 | 5.88 |
| Romance | 87.27 | 3.64 | 9.09 |
| Crime | 87.04 | 7.41 | 5.56 |
| Horror | 86.79 | 3.77 | 9.43 |
| Sport | 84.09 | 4.55 | 11.36 |
| Thriller | 83.33 | 1.85 | 14.81 |
| Mystery | 83.02 | 3.77 | 13.21 |
| Music | 81.63 | 16.33 | 2.04 |
| Animation | 80.77 | 11.54 | 7.69 |
| War | 80.39 | 5.88 | 13.73 |
| Family | 80.00 | 7.27 | 12.73 |
| Comedy | 79.25 | 16.98 | 3.77 |
| Drama | 77.78 | 14.81 | 7.41 |
| Action | 72.55 | 15.69 | 11.76 |

## C IMPLEMENTATION DETAILS ON COMPARATIVE ALGORITHMS

### C.1 PROBLEM TRANSFORMATION AND SPECIALIZED ALGORITHMS

All PT and SA algorithms are implemented using PyLDL [12] Python Package. Since these methods are not designed to leverage temporal features, we apply average pooling to aggregate audio and visual inputs before feeding them into the models. For LDL-LRR Jia et al. (2019) and TLRLDL Kou12 et al. (2024), as recommended, we apply StandardScaler feature preprocessing and perform hyperparameter tuning on weight of different loss and regularization terms on the validation set.

### C.2 ADAPTED ALGORITHMS

**CubeMLP** Sun et al. (2022) combines temporally aligned unimodal features and mixes them across time, feature, and modality dimension using a structure consisting of 3 independent MLP units.

---

[12]https://github.com/SpriteMisaka/PyLDL/tree/main

Table 11: Performance across different emotion categories.

| Emotion Category | % Correct | % Not Sure | % Incorrect |
|---|---|---|---|
| annoyance | 100.00 | 0.00 | 0.00 |
| relief | 100.00 | 0.00 | 0.00 |
| confusion | 94.64 | 5.36 | 0.00 |
| approval | 93.94 | 0.00 | 6.06 |
| curiosity | 92.31 | 0.00 | 7.69 |
| amusement | 91.53 | 5.29 | 3.17 |
| admiration | 90.41 | 5.17 | 4.43 |
| surprise | 85.71 | 2.86 | 11.43 |
| disapproval | 84.71 | 7.85 | 7.44 |
| excitement | 82.35 | 17.65 | 0.00 |
| disappointment | 73.91 | 8.70 | 17.39 |
| disgust | 72.73 | 9.09 | 18.18 |
| sadness | 66.67 | 20.00 | 13.33 |
| fear | 66.67 | 11.67 | 21.67 |
| realization | 66.67 | 0.00 | 33.33 |
| joy | 50.00 | 16.67 | 33.33 |
| caring | 50.00 | 25.00 | 25.00 |
| grief | 50.00 | 16.67 | 33.33 |
| anger | 0.00 | 75.00 | 25.00 |
| embarrassment | 0.00 | 33.33 | 66.67 |
| nervousness | 0.00 | 0.00 | 100.00 |

**CTEN** Zhang et al. (2023) incorporates both uni-modal and cross-modal temporal attention mechanisms over visual and audio features extracted from key snippets, and further introduces an erasing strategy to localize context- and audio-relevant information in a weakly supervised setting. The major modification is that we only forward the processed visual and audio features to the model, so the ResNet encoders in the original CTEN to process raw modalities are removed. Further, since the erasing module requires access to the full original video, which is unavailable for Video2Reaction, we report benchmark results using CTEN without the erasing component. For the applicable hyperparamters, we follow the default values proposed in Zhang et al. (2023) to construct the model, and report them in Table 12.

**MMIM** Han et al. (2021) introduces a two-stage end-to-end pipeline that learns fused representations from refined cross-modal features. The model is trained jointly to optimize both downstream task performance and mutual information between the fused representation and the unimodal input. In the first stage, mutual information is estimated by modeling a Gaussian mixture over positive and negative group. To assign samples to either group on Video2Reaction, we compute the summed probabilities of all positive and negative emotions for each sample, assigning it to the group with the dominant emotion category. Since the inputs are processed latent features, we replace the RNN models in the original MMIM with simple fully connected layers to align the latent dimension. For the applicable hyperparamters, we follow the default values proposed in Han et al. (2021) to construct the model, and report them in Table 12.

For the three models, we use Pytorch with 1 GPU use the same schedule to train them for better performance comparison. We use cross entropy loss between the output logits and the ground truth label distributions as the training loss, use SGD as the optimizer, and use StepLR as the optimizer scheduler. Details on the hyperparameters are reported in Table 13. Additionally, when training MMIM, it requires 2 separate training stages, and we use the same set of hyperparameters in Table 13 for the two optimizers and schedulers in the two stages. Also, when training MMIM, it uses a single likelihood maximization loss in the first stage, and requires extra Contrastive Predictive Coding score and Gaussian mixture based mutual information estimation in the loss, besides the regular cross entropy loss we apply to every model.

Table 12: Model-specific Hyperparameters for Algorithm Adaptation (AA)

| Model | Hyperparameter |
|---|---|
| CubeMLP | encoders=lstm
d_common=128
activate=gelu
time_len=16
d_hiddens=[[16, 2, 128],[8, 2, 128],[4, 1, 128]]
d_outs = [[2, 2, 128],[2, 2, 128],[2, 2, 2]]
dropout=0.1
features_compose_t="cat"
features_compose_k="cat" |
| CTEN | n_classes=21
seq_len=16
audio_embed_size=1024
visual_embed_size=768 |
| MMIM | alpha=0.1
beta=0.1
update_batch=1
clip=1.0
dropout_a=0.1
dropout_v=0.1
dropout_prj=0.1
n_layer=1
cpc_layers=1
d_vout=16
d_aout=16
d_tout=16
d_tfeatdim=768
d_afeatdim=1024
d_vfeatdim=768
n_class=21
d_prjh=128
pretrain_emb=768
mmilb_mid_activation=relu
mmilb_last_activation=tanh
cpc_activation=tanh |

Table 13: Adapted Algorithms Training Hyperparameters

| Hyperparam | Value |
|---|---|
| Learning rate | 1e-3 |
| Weight decay | 1e-4 |
| Momentum | 0.9 |
| #Epochs | 200 |
| Batch size | 128 |
| StepLR step size | 50 |
| StepLR gamma | 0.5 |

## C.3 FOUNDATION VISION-LARGE LANGUAGE MODELS

To extract the model's estimated probability of each reaction, we prompt with multiple-choice questions and use the next token's probability as the predicted probability of each reaction category. The prompt is provided below:

---

**Prompt for Audience Reaction Forecasting using Foundation VLMs**

**User:**
You are an AI assistant. Your task is to forecast potential audience reactions to a movie scene. You will be provided with the video and some video context.
`[Video Input]`
In this scene, `<clip description>`.
Given this clip, what do you think a viewer's reaction would be?
Choose the letter of the most appropriate reaction. ¡List of 21 choices are presented line by line here¿

**Assistant:**
Based on the scene, a viewer's reaction would be letter `<your choice here>`.

---

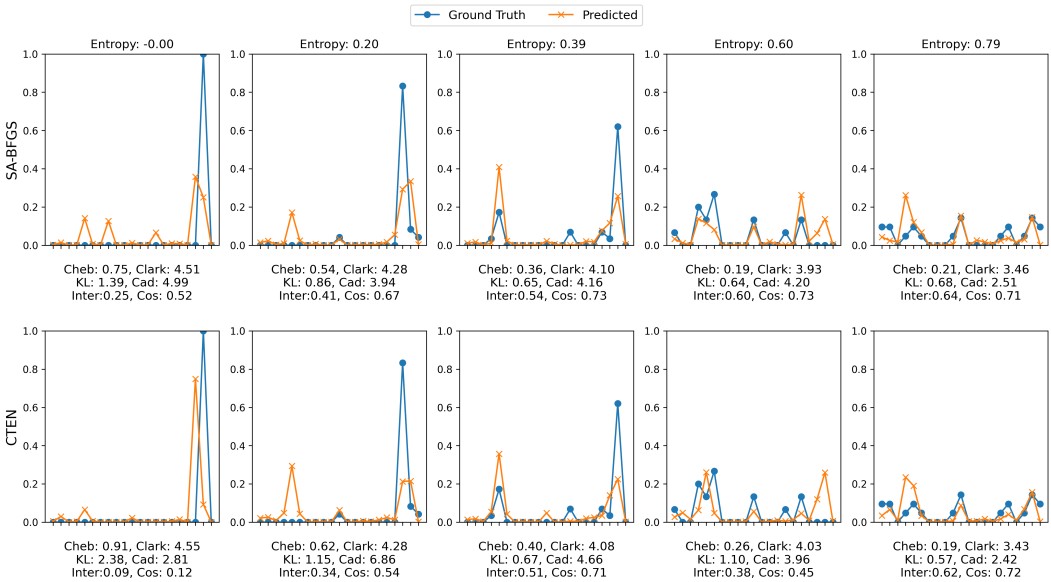

Figure 4: Visual Comparison of Predicted Reaction Distribution of Two Leading Baseline Algorithms. One random sample is selected for each bin of groundtruth distribution entropy.

# D   ADDITIONAL BENCHMARK RESULTS

## D.1   VISUALIZATION OF PREDICTED DISTRIBUTION VS. GROUNDTRUTH DISTRIBUTION

Figure 4 illustrates how the two leading algorithms in our benchmark (SA-BFGS and CTEN) model groundtruth label distribution across different entropy levels. Lower entropy represents more unimodal distribution and higher entropy represents more uniform distribution.

## D.2   LONG-TAIL CHALLENGE IN DOMINANT REACTION EVALUATION

Table 14 summarizes the number of videos with certain reaction showing up in top 3 of the groundtruth distribution as well as F1 score of each class.

# E   INSTRUCTION FOR HUMAN QUALITY REVIEW OF AUTOMATIC ANNOTATIONS

To assess the quality of automated reaction annotation, we randomly sample 100 movie clips with balanced representation across all movie genres. From each clip, 10 comments are randomly selected, yielding a total of 1,000 comments for human evaluation. Due to the subjective nature of fine-grained audience reactions, each comment is independently reviewed by two annotators. In cases of disagreement, a third annotator is consulted, and the final label is determined by majority vote. In total, five annotators participated in this quality review process. Instruction for the annotation is described below.

Table 14: Reaction Class, Number of Videos with Reaction in Top 3, and Top-3 F1 Score of SA-BFGS algorithm

| Reaction Class | Number of Videos in Top 3 | $F1_3$ Score |
|---|---|---|
| Sadness | 145 | 0.3058 |
| Disgust | 179 | 0.2482 |
| Grief | 39 | 0.1852 |
| Fear | 290 | 0.5509 |
| Disapproval | 1472 | 0.8283 |
| Disappointment | 264 | 0.1590 |
| Embarrassment | 6 | - |
| Nervousness | 8 | - |
| Annoyance | 2 | - |
| Anger | 18 | 0.0952 |
| Confusion | 314 | 0.1663 |
| Realization | 8 | 0.1667 |
| Caring | 14 | 0.1176 |
| Curiosity | 33 | - |
| Relief | 10 | - |
| Approval | 136 | 0.1435 |
| Surprise | 285 | 0.0471 |
| Excitement | 118 | 0.0714 |
| Amusement | 1161 | 0.7540 |
| Admiration | 1612 | 0.8760 |
| Joy | 96 | 0.1228 |

**Instruction for Human Reviewers**

The LLM agent was tasked with annotating the fine-grained reaction of the YouTube commenter towards the movie clip.

For each comment, choose one of the following ratings:

- **Correct**: You think there is a high chance that the LLM response is correct. Sometimes the model may stretch to infer the audience's intent, but if you believe the guess is reasonable, rate it as Correct.

- **Incorrect**: You think there is a low chance that the LLM response is correct. If the model's guess seems too far-fetched or if a different reaction label clearly fits better, rate it as Incorrect.

- **Not Sure**: You cannot understand the comment well enough to make a judgment.

