# OpenReview forum: "Video2Reaction: Mapping Video to Audience Reaction Distribution in the Wild"
_ICLR.cc/2026/Conference — Submitted to ICLR 2026_

### Official Review · Reviewer_AxqW · 2025-10-26

**Soundness:** 3
**Presentation:** 4
**Contribution:** 3
**Rating:** 4
**Confidence:** 5

**Summary:**

In this work, the authors pointed out a critical gap in affective computing research—the limited attention to induced emotion recognition, which focuses on understanding how audiences emotionally respond to video content, rather than the perceived emotions conveyed by characters or intended by filmmakers. Shifting the emphasis from the creator’s intent to the viewer’s reaction is essential for advancing content creation, media analysis, and personalized recommendation. To address this gap, the authors introduce Video2Reaction, a large-scale multimodal dataset comprising over 10,000 movie clips, each annotated with audience reaction distributions derived from real-world viewer comments. Unlike prior datasets that rely on single-label or perceived emotion annotations, Video2Reaction is annotated using the distribution of emotional responses, capturing the diversity and nuance of audience reactions. The authors further benchmark a wide range of existing methods to evaluate their strengths and limitations on this new task.

**Strengths:**

1. The authors provide a comprehensive and well-documented description of the experimental setup and protocol design. The experiments encompass a wide range of methods, effectively establishing strong and diverse baselines.
2. The benchmark is described in a thorough and well-organized manner, making it easy for readers to follow
3.  The figures and tables are clear

**Weaknesses:**

1. A key concern lies in the reliability of the emotion ground truth. The authors note that many viewer comments may reflect personal attitudes toward actors or filmmakers rather than genuine emotional responses to the video content. To mitigate this issue, they employ prompt-based filtering with minimal human intervention. Since this step is critical to preserving the purity and validity of the dataset, it would be beneficial to provide an evaluation of the filtering process to demonstrate its effectiveness. Moreover, the authors should include evidence that the employed agent models are capable of accurately extracting fine-grained emotional cues from comments, as this capability is a core assumption underlying the reliability of the annotations. However, based on the results in Table 6, the foundation VLM appears to lack the ability to capture fine-grained emotions; therefore, it remains unclear why the LLM would possess this capability merely when the video caption is provided.

2. The authors propose to use emotion distributions for comparison; however, it would be beneficial to clearly demonstrate the advantages of this distribution-based approach over traditional single-label annotation. According to the paper, only up to three human annotations are used to assess label validity, which appears insufficient to ensure the reliability of the annotations. Moreover, for the emotion distribution itself, it remains unclear how the proportion of each emotion is determined and validated—this aspect needs further clarification and justification.

3. It would be helpful to include a more explicit comparison between this work and related studies [1]. That prior work also targets viewer-centered emotion recognition, using a combination of three emotions as ground-truth labels to account for emotional mixtures, with video sources collected from social media. In comparison, the current dataset is derived from films, which are typically more scripted and contextually constrained. From this perspective, the previous work may better align with the notion of “in-the-wild” data, as it includes more spontaneous content such as dashcam footage.

4. Another potential concern is the bias in viewer comments, as not everyone is equally likely to leave a comment. Those who do are often younger or more extroverted individuals, which may skew the emotional responses captured in the dataset. Consequently, the collected emotions may primarily reflect the reactions of this particular group of people rather than representing the broader audience.

5. For highly intense emotions such as anger or fear, the emotional trigger often occurs within a very short time frame. It would be helpful to clarify whether representing each scene with a single frame is sufficient to capture such transient emotional expressions.

6. The authors use clip descriptions provided by the channel, which are likely to be general summaries lacking detailed information about specific scenes or objects within each frame. The absence of these critical triggers in the captions makes it difficult to establish a reliable connection between human emotional reactions and the actual video content.


[1] Mazeika, M., Tang, E., Zou, A., Basart, S., Chan, J. S., Song, D., ... & Hendrycks, D. (2022). How would the viewer feel? Estimating wellbeing from video scenarios. Advances in Neural Information Processing Systems, 35, 18571-18585.

**Questions:**

None

---

> ### Author Response · Authors · 2025-11-21
> **(1/2) Initial Response**
>
> We thank the reviewer for recognizing our comprehensive benchmark and leaving helpful feedback. We would like to provide some clarifications on the concerns and questions the reviewer pointed out.
>
> **1\. \[experiments\] the authors should include evidence that the employed agent models are capable of accurately extracting fine-grained emotional cues from comments, as this capability is a core assumption underlying the reliability of the annotations.**
> Regarding the reliability of the emotion ground truth, we would like to refer to our human evaluation results in the \*\*Quality of the reaction annotation\*\* paragraph (Section 3.3, page 5), Appendix B.2 Table 9 (summary of human evaluation) and Table 10, 11 (error analysis breakdown by movie genres and emotion categories). The results suggest that overall our annotation pipeline, an ensemble of 3 LLM agents, can extract fine-grained emotions with an accuracy of 86%. The LLM-based annotation pipeline performs reasonably well for most common emotions like annoyance and approval, but falls short for underrepresented emotions like nervousness and embarrassment.
>
> **2\. \[clarification\] based on the results in Table 6, the foundation VLM appears to lack the ability to capture fine-grained emotions; therefore, it remains unclear why the LLM would possess this capability merely when the video caption is provided.**
> We use LLMs and VLMs for two different tasks. LLMs are used during data curation to label emotions per comment (text), based on the comment text and the short description text of the targeted movie clip. The emotion annotation capabilities of LLMs for text have been well studied and leveraged in recent studies \[1\]. On the other hand, VLMs serve as a benchmark for predicting the reaction distribution on the curated dataset, by taking the embedding video as input and outputting a distribution over 21 emotion categories, which is a much harder and under-studied task. The low performance of VLMs on the downstream task of detecting emotion distributions per movie clip is irrelevant to the upstream dataset curation process, thus one may not connect the poor performance of VLMs to the ability of LLMs emotion detection capacity.
> \[1\] Niu, M., Jaiswal, M., & Provost, E. M. (2024). From text to emotion: Unveiling the emotion annotation capabilities of llms. INTERSPEECH.
>
> **3\. \[discussion\] it would be beneficial to clearly demonstrate the advantages of this distribution-based approach over traditional single-label annotation.**
> As demonstrated in Figure 3(b) (page 6), there is a wide range in the probability of the dominant reaction across videos and the median probability of the dominant reaction per video is about 0.4. This suggests that if we only use the dominant reaction as the single label, we will fail to capture the mixed induced emotions to controversial videos.
>
> **4\. \[clarification\] for the emotion distribution itself, it remains unclear how the proportion of each emotion is determined and validated**
> The annotations are done at the comment level and then aggregated at the clip level to construct an emotion distribution at clip level (for example, a clip with 70 comments that are annotated as “surprise” and 30 comments annotated as “annoyance” will construct an emotion distribution with 0.70 surprise and 0.3 annoyance). The validation is done at the comment level for a data subset of 1000 comments drawing from clips across movie genres.

---

> ### Author Response · Authors · 2025-11-21
> **(2/2) Initial Response**
>
> **5\. \[clarification\] For highly intense emotions such as anger or fear, the emotional trigger often occurs within a very short time frame. It would be helpful to clarify whether representing each scene with a single frame is sufficient to capture such transient emotional expressions.**
> The scene detection is conducted in an adaptive manner with a very low threshold so for a fast-paced plot with a lot of visual changes in a short time frame, it would be split into multiple scenes. We will add this implementation detail to the Appendix.
>
> **6\. \[methodology\] The authors use clip descriptions provided by the channel, which are likely to be general summaries lacking detailed information about specific scenes or objects within each frame. The absence of these critical triggers in the captions makes it difficult to establish a reliable connection between human emotional reactions and the actual video content.**
> We pass frame-level visual information to the models which already includes scene and object information. We agree with the reviewer that there is a lot of room to improve our task’s baseline performance by enriching current features with a more specific clip description (for example, one generated by a video understanding model). We hope that by presenting this novel and practical task and a comprehensive benchmark, we can encourage researchers to build upon our work to improve upon our baseline performance.
>
> **7\. \[methodology\] Another potential concern is the bias in viewer comments, as not everyone is equally likely to leave a comment.**
> We acknowledge that, like any other social media-based data collection effort, our dataset is subject to selection bias. However, given the global usage and high engagement of YouTube as one of the world’s largest entertainment platforms, we believe that analyzing audience reactions on YouTube captures an important and sizable portion of real-world viewer responses. Additionally, from the pragmatic standpoint of content creators, their primary interest is in understanding the reactions of users who actively engage with their content, as this group provides the most informative and actionable feedback.
>
> During the design phase, we considered an alternative approach of recruiting participants to watch videos and report their reactions as in prior studies \[1\]. While that approach is valuable, they also introduce their own confounding factors (e.g., small sample sizes, demographic homogeneity, study-participant effects) and cannot approximate the scale or diversity of naturally occurring audience reactions. Given that audience engagement is an application that requires frequent updates, we have aimed to strike a careful balance between data quality, cost, and sustainability while developing the data collection pipeline, with the goal of enabling iterative improvements to the benchmark.
>
> \[1\] Yoann Baveye, Emmanuel Dellandrea, Christel Chamaret, and Liming Chen. Liris-accede: A video database for affective content analysis. IEEE Transactions on Affective Computing, 6(1):43–55, 2015\.
>
> **8\. \[related work\]  It would be helpful to include a more explicit comparison between this work and related studies \[1\].**
> This is a great addition to our related works and further supports the need to model induced emotion as a distribution rather than a single label. We will add this dataset to Table 1 and add a separate paragraph discussing the connection and difference between our work and their Video Cognitive Empathy (VCE) dataset \[1\] as explained below.
>
> First of all, we would like to clarify that our use of the term “in the wild” refers to viewers experiencing the content in uncontrolled, self-selected settings (i.e., watching at their own time and environment), not to the nature of the content itself. Therefore, our dataset is done in the wild setting while VCE \[1\] is not. Secondly, while \[1\] derives emotion distributions from 12–15 annotators drawn solely from the U.S., our dataset aggregates reactions from 10–100 viewers per clip across a global population. Moreover, in addition to the top-k evaluation used in \[1\], our benchmark incorporates a broader set of metrics designed to capture diverse error types, which is crucial given that the utility of this task depends heavily on the fidelity of the predicted emotion distribution (see Table 4). Lastly, our dataset includes two types of audio features which are important for movie content while the VCE dataset only includes visual features.

---

> > ### Comment · Reviewer_AxqW · 2025-11-24
> >
> > Thank you for the comments, which have helped me gain a better understanding of the paper. However, I still have several concerns:
> >
> > 1. According to the results in Table 11, several emotion categories exhibit low accuracy. Given their unreliability, why are these emotions still incorporated into the analysis?
> >
> > 2. As mentioned in the referenced work, while LLMs possess emotion annotation capabilities, their performance degrades as the number of emotion classes increases—for example, GPT-4 achieves only 0.375 F1-score on GoEmotions with 27 categories. This suggests that although feasible, the annotation reliability is limited.
> >
> > 3. If single dominant emotion prediction is unstable, would using the top-3 predicted emotions be more robust? Incorporating full emotion distribution introduces challenges, particularly in accurately estimating the proportion of each emotion. Additionally, three annotators were used to validate the distribution—how is this distribution evaluated. From the ta,sk's practical aspect,  why is precise proportional estimation necessary?
> >
> > 4. It would be beneficial to provide the validation details of frame selection in the Appendix to further support the reliability of the approach, since the edits are available during the rebuttal.
> >
> > 5. Regarding the definition of “in-the-wild,” what specifically constitutes a “constrained viewer” in this context?
> >
> > Thanks.

---

> > > ### Author Response · Authors · 2025-11-26
> > >
> > > Thank you for your response. We are glad our responses help you better understand the paper. We would like to address your additional concerns below.
> > >
> > > > According to the results in Table 11, several emotion categories exhibit low accuracy. Given their unreliability, why are these emotions still incorporated into the analysis?
> > >
> > > Similar to the GoEmotions paper, our criteria to include or exclude emotion categories depend on what best represents the data, not on the annotation quality (accuracy for our LLM-annotated pipeline or inter-rater correlation for human review). Excluding emotion categories that are more challenging to extract correctly risks missing important audience reactions. However, we agree that this unreliability is a problem. In the future iteration of the pipeline, we plan to revise the pipeline to have human in the loop for a small set of challenging categories. What do you think about that idea? If you have any other suggestions, we would be happy to consider them as well.
> > >
> > > > As mentioned in the referenced work, while LLMs possess emotion annotation capabilities, their performance degrades as the number of emotion classes increases—for example, GPT-4 achieves only 0.375 F1-score on GoEmotions with 27 categories. This suggests that although feasible, the annotation reliability is limited.
> > >
> > > The paper reports macro F1 score which is a non-weighted average of all class-wise F1 score. Therefore, for a taxonomy of 27 categories, a macro F1 score of $0.375$ is actually not very low. Additionally, the referenced paper further shows that evaluation metrics based on human annotations tend to underestimate the performance of GPT-4. Additional human evaluation comparing LLM-generated labels and human-annotated labels demonstrates that reviewers preferred GPT-4’s labels on 71.1% of the samples in the GoEmotions dataset, indicating that LLM annotations can be more reliable than conventional human annotations.
> > >
> > > > If single dominant emotion prediction is unstable, would using the top-3 predicted emotions be more robust? Incorporating full emotion distribution introduces challenges, particularly in accurately estimating the proportion of each emotion. Additionally, three annotators were used to validate the distribution—how is this distribution evaluated. From the task's practical aspect, why is precise proportional estimation necessary?
> > >
> > > In the Dominant Reaction Evaluation metrics, we do include the top-3 weighted F1 score (Table 6).However, not all downstream tasks require only identifying the top emotions. Some tasks depend on accurately estimating the proportions in the full distribution. For example, in video editing applications, a creator may want to make minimal edits that selectively increase the intensity of a specific emotion. If the model only outputs the top-3 emotions without proportional information, such fine-grained adjustments would not be possible. This is why we evaluate both dominant-reaction metrics and full-distribution metrics.
> > >
> > > > It would be beneficial to provide the validation details of frame selection in the Appendix to further support the reliability of the approach, since the edits are available during the rebuttal.
> > >
> > > We added more details of scene detection and frame selection in the Appendix A.3.
> > >
> > > > Regarding the definition of “in-the-wild,” what specifically constitutes a “constrained viewer” in this context?
> > >
> > > A “constrained viewer” refers to a viewer who does not voluntarily choose to watch and react to a piece of multimedia content. In other words, their viewing experience is induced by the experimental setup, such as being recruited to watch a video for annotation, rather than occurring naturally in their own context, time, and environment.

---

### Official Review · Reviewer_vNMM · 2025-10-30

**Soundness:** 2
**Presentation:** 2
**Contribution:** 2
**Rating:** 4
**Confidence:** 4

**Summary:**

The paper introduces Video2Reaction, a dataset of 10,348 movie clips paired with audience reaction distributions derived from YouTube comments. The authors claim this is the first dataset capturing distributional induced emotions from cinematic content "in the wild." They use a two-stage LLM-based pipeline to annotate comments and benchmark various Label Distribution Learning (LDL) algorithms on the task of predicting reaction distributions from video features alone.

**Strengths:**

Large scale dataset with 400 hours of video and 800K comments - One of the largest multimodal emotion datasets to date, providing substantial data for model training

Addresses the distinction between perceived and induced emotions

Comprehensive benchmarking across multiple algorithm categories. The evaluation spans PT, SA, AA, and foundation VLMs and provides useful baseline comparisons

Two-stage annotation pipeline with multi-agent voting that is highly scalable and could support future dataset expansion

Provides both full distribution and dominant reaction evaluation metrics

The paper is clearly written with well-structured sections and informative figures.

**Weaknesses:**

The Validation Protocol for the LLM pipeline is not necessarily convincing. The human verification shows LLM predictions to reviewers before asking if they're "correct," possibly introducing confirmation bias. The reported 86% accuracy is meaningless under this flawed protocol.

The paper conflates "audience reactions" with "YouTube commenter reactions." Commenters possibly represent only 1-5% of viewers and are possibly also self-selected for stronger emotional engagement. The dataset therefore doesn't measure exactly what it claims to measure. (Thought they somewhat address it in the limitations, this could have been discussed more prominently.)

The related work section could have better contextualized existing datasets that would enable validation of the annotation pipeline.

Lack of validation on existing emotion–comment datasets: No attempt to validate the LLM-based annotation pipeline on established datasets (e.g., Cong Xu, Lu Liu, Liang Jin, Guoguang Du, Zhenhua Guo, Yaqian Zhao, Xuanjing Huang, Rengang Li, et al. Infer induced sentiment of comment response to video: A new task, dataset and baseline. Advances in Neural Information Processing Systems, 37:103737–103750, 2024. ) where ground truth emotion labels for comments already exist.

Although the dataset is large and thematically somewhat relevant, the methodological flaws (biased validation and unclear representativeness) limit its usefulness for future research.

**Questions:**

Why use a validation protocol that exposes reviewers to LLM predictions rather than blind annotation?
How do you justify treating YouTube commenters as representative of the general viewing audience?
Why was the annotation pipeline not validated on existing comment–emotion datasets?
How do you account for differences in comment count? Was there an artificial cap of 100 comments? (Max value is 100 in table 3a)
How did you handle your dataset splits given the dataset imbalance?

**Details Of Ethics Concerns:**

Justify whether humans should be informed and compensated.

---

> ### Author Response · Authors · 2025-11-21
> **(1/2) Initial Response**
>
> We thank the reviewer for recognizing our key contributions in providing a large-scale dataset on induced emotions, which is understudied compared to perceived emotions, and in building a comprehensive benchmark with both distribution and dominant reaction evaluation. We would like to address each of the reviewer’s concerns and questions below.
> **1\. \[methodology\] The paper conflates "audience reactions" with "YouTube commenter reactions." Commenters possibly represent only 1-5% of viewers and are possibly also self-selected for stronger emotional engagement.**
> We do not claim that social-media-derived reactions should be the only metric for audience reaction modeling, but we argue that they are undeniably one of the key metrics and offer complementary insights to recruited viewers’ interviews and surveys. Given the global usage and high engagement of YouTube as one of the world’s largest entertainment platforms, we believe that analyzing audience reactions on YouTube captures an important and sizable portion of real-world viewer responses. Additionally, from the pragmatic standpoint of content creators, their primary interest is in understanding the reactions of users who actively engage with their content, as this group provides the most informative and actionable feedback.
>
> We initially considered an alternative which is to recruit a small set of participants to watch and rate the videos. While that approach is valuable, they also introduce their own confounding factors (e.g., small sample sizes, demographic homogeneity, study-participant effects). Our dataset aims to complement existing efforts by providing a representation of audience reaction that would be difficult to obtain through small-scale annotation alone.
>
> **2\. \[experiments\] No attempt to validate the LLM-based annotation pipeline on established datasets (e.g., Cong Xu, Lu Liu, Liang Jin, Guoguang Du, Zhenhua Guo, Yaqian Zhao, Xuanjing Huang, Rengang Li, et al. Infer induced sentiment of comment response to video: A new task, dataset and baseline. Advances in Neural Information Processing Systems, 37:103737–103750, 2024\. ) where ground truth emotion labels for comments already exist.**
>
> This is a great idea. We looked into this to see whether we can run this experiment. Our annotation pipeline requires a clip description to set as context to understand the emotion expressed in comments. Unfortunately, this dataset does not come with a clip description or a raw video to generate the corresponding clip description using VLMs. We have reached out to the authors to see whether we can obtain clip descriptions or raw video. If we can get the data in time, we will run our LLM annotation pipeline and get the evaluation.
>
> However, we would like to emphasize that we took great efforts to ensure and evaluate the quality of the LLM-based annotation pipeline. Two human reviewers independently review the labels on a substantial test set of 1,000 comments. Based on our prior experience with emotion annotation, we know that genre can impact difficulty, so we construct the test set to ensure good representation across genres. All four human reviewers involved in this process are undergraduate or graduate students with prior experience in affective computing.
>
> **3\. \[methodology\] Why use a validation protocol that exposes reviewers to LLM predictions rather than blind annotation?**
> We chose verification over direct human annotation to avoid the substantial annotation noise reported in prior work on fine-grained emotion classification (where most categories have interrater correlations \<0.5 \[1\] ). In the initial phase of dataset construction, we recruited two annotators to provide multi-label annotations for a set of 330 comments. However, they highlighted the difficulty of choosing from 21 emotion categories, exceeding the typical human limit of around 7 items \[2\], which led to noisy labels and a bias toward common emotions like happiness and sadness. Multi-label annotation also proves time-intensive, making it impractical to scale to a larger size of test set within budget. As a result, we adopted a human verification approach, using double independent verification to reduce label noise without requiring full manual annotation.
>
> \[1\] Demszky, D., Movshovitz-Attias, D., Ko, J., Cowen, A., Nemade, G., & Ravi, S. (2020). GoEmotions: A dataset of fine-grained emotions. *arXiv preprint arXiv:2005.00547*.
> \[2\] Miller, G. A. (1956). The magical number seven, plus or minus two: Some limits on our capacity for processing information. Psychological review.

---

> ### Author Response · Authors · 2025-11-21
> **(2/2) Initial Response**
>
> **4\. \[clarification\] How do you account for differences in comment count? Was there an artificial cap of 100 comments? (Max value is 100 in table 3a)**
> We use the percentage of comments with each reaction to construct a normalized distribution. For scalability, we use the top 100 comments (sorted by popularity). This cap is imposed primarily for computational efficiency, while still ensuring that we capture the most representative and widely engaged audience reactions. We will add this clarification to the revision of the manuscript.
>
> **5\. \[clarification\] How did you handle your dataset splits given the dataset imbalance?**
> The dataset is split into train, validation, and test set by using the ratio 0.7 / 0.1 / 0.2 and by ensuring that (1) the distribution of the dominant reaction in all splits are roughly similar and (2) the train set has all the reactions present in the dataset. We will add this detail to the revised version of the manuscript.

---

> > ### Comment · Reviewer_vNMM · 2025-11-25
> > **thanks for the responses**
> >
> > Thank you for the detailed response.
> > Regarding the verification, I understand the reasoning behind adopting the verification protocol given feasibility constraints, but it is still not convincing that this provides sufficient validation for the label quality itself.
> >
> > In this respect, I am aligned with Reviewer tQDy's suggestions for better validation. As they propose: showing that "the model's agreement with humans is similar to human agreement among themselves in the open label setup" would provide stronger evidence than your current verification protocol. Furthermore, if humans genuinely can't perform this task well, you could "assign random emotion labels to comments and ask humans to verify them" to quantify the acquiescence bias in your protocol.
> >
> > Additionally, obtaining clip descriptions for the CMSV dataset, would be very valuable for the benchmark evaluation.
> >
> > A few other points:
> > The method section should explicitly state that the pipeline requires clip descriptions. Currently this can only be inferred from the prompts. Was the pipeline ablated without clip descriptions to show that they are necessary?
> >
> > After this discussion, the paper's core contribution is confusing to me: Is it the benchmark about generating insights for content creators or actually modeling induced emotions?
> > The pivot between scientific framing (induced emotions) and pragmatic justification (creator analytics) makes the paper's purpose unclear.

---

> > > ### Author Response · Authors · 2025-11-26
> > >
> > > Thank you for engaging with us to improve the manuscript. We would like to address your follow-up questions and concern below.
> > >
> > > > if humans genuinely can't perform this task well, you could "assign random emotion labels to comments and ask humans to verify them" to quantify the acquiescence bias in your protocol.
> > >
> > > We just completed this test on a random sample of 100 comments. We presented the reviewer with two options, one is LLM-annotated label and one is randomly assigned labels, and the reviewer is not aware which one is which. The result below shows that (1) human verification approach can distinguish between correct and incorrect labels and not subject to confirmation bias and (2) LLM-annotated labels are preferred in most cases.
> > >
> > > | Category              | Value  |
> > > |-----------------------|--------|
> > > | Total Responses       | 100    |
> > > | LLM Preference %      | 87.00% |
> > > | Random Preference %   | 2.00%  |
> > > | Neither Preferred %   | 11.00% |
> > >
> > > > Additionally, obtaining clip descriptions for the CMSV dataset, would be very valuable for the benchmark evaluation.
> > >
> > > We agree that it would be a good addition to validate the quality of our annotation pipeline. We finally got the links of the Tiktok videos used in the CMSV dataset so that we can download the videos. Since Tiktok videos do not come with informative descriptions, we will use a VLM to generate a text description of the clip. We are in the process of downloading videos. Though we are unsure whether we can complete this experiment in time for the rebuttal, we will still give this an attempt to improve the next version of our manuscript. Thank you for this suggestion.
> > >
> > > > The method section should explicitly state that the pipeline requires clip descriptions. Currently this can only be inferred from the prompts. Was the pipeline ablated without clip descriptions to show that they are necessary
> > >
> > > I added the requirement of clip descriptions for the LLM annotation pipeline in Appendix B.1 (L345-355). This design decision stems from our qualitative results of the first iteration of the LLM annotation pipeline. For about half of the comments we manually checked, it is difficult to extract the emotion without the context of the clip.
> > >
> > > > After this discussion, the paper's core contribution is confusing to me: Is it the benchmark about generating insights for content creators or actually modeling induced emotions? The pivot between scientific framing (induced emotions) and pragmatic justification (creator analytics) makes the paper's purpose unclear.
> > >
> > > The paper aims to model / predict induced emotions which will generate insights for content creators. We believe that our paper both advances scientific understanding of how to model induced emotion distributions with different distribution learning algorithms and enables content creators to forecast audience reaction using these algorithms.

---

### Official Review · Reviewer_3vdS · 2025-10-31

**Soundness:** 3
**Presentation:** 3
**Contribution:** 3
**Rating:** 6
**Confidence:** 2

**Summary:**

The paper introduces Video2Reaction, a large-scale multimodal benchmark that maps movie clips to the distribution of audience emotions expressed in social-media comments. Unlike traditional perceived-emotion datasets, it focuses on induced emotions , what viewers feel rather than what characters portray. It contains over 10k clips (400 h) from 1.5 k movies with 800 k comments, annotated through a two-stage multi-agent LLM pipeline and verified by humans. The task is formulated as label-distribution learning (LDL), and the benchmark compares classical LDL, multimodal adaptations, and foundation vision-language models. Results show that specialized LDL algorithms (eg SA-BFGS) perform best, while VLMs lag behind in zero-shot settings.

**Strengths:**

Largest known dataset for induced audience emotion in the wild.
Thoughtful two-stage LLM annotation pipeline with partial human auditing.
Comprehensive benchmark with multiple algorithm classes and clear metrics.
Highlights meaningful gaps between specialized LDL models and general-purpose VLMs.

**Weaknesses:**

Annotation reliability and bias analysis are thin; LLM bias or sarcasm mislabeling could skew distributions.
Dataset remains culturally and platform-biased (YouTube only).
Results show moderate accuracy; limited insight into failure cases.
Contribution is more engineering than conceptual. Some claims about 'first of its kind' feel overstated given existing induced-emotion datasets (eg CMSV 2024).

**Questions:**

How does the LLM ensemble handle cross-lingual comments?
Can future updates dynamically re-label older clips to track temporal drift?
Were annotators or models ever confused by sarcasm or meme responses?
Could foundation models be fine-tuned efficiently using this dataset?

---

> ### Author Response · Authors · 2025-11-21
> **(1/2) Initial Response**
>
> We thank the reviewer for acknowledging our two-stage annotation pipeline and the comprehensive benchmark for this new task of predicting distributions of audience reaction in the wild. We would like to respond to each of your comments/questions below.
>
> **1\. \[experiments\] Annotation reliability and bias analysis are thin**
> We provide additional error analysis of the annotation pipeline in Appendix B.2 (Table 10 and 11), which shows that our annotation pipeline performs reasonably well across movie genres but carries a clear difference in performance across emotion categories. It performs well for the most common emotions like annoyance, approval but badly for underrepresented emotions like anger, embarrassment, and nervousness.
>
> **2\. \[methodology\] Dataset remains culturally and platform-biased (YouTube only)**
>
> We agree with the reviewer that one of the limitations of our dataset is that it is currently based on audience reaction data from a single platform (as mentioned in the Limitations section). However, Youtube is one of the biggest platforms worldwide from which we can extract global audience reaction so we think it is a good representation of audience reaction. In the future release, with our scalable annotation pipeline, we will attempt to extend it to further platforms.
>
> **3\. \[discussion\] Contribution is more engineering than conceptual**
> We would like to emphasize that designing a meaningful and practical benchmark to discover a video clip’s audience reaction requires substantial research efforts rather than merely engineering. First of all, the decision to represent audience reaction as a distribution, instead of a single label, is a conceptual choice grounded in the inherently multi-faceted and diverse nature of emotional response. Second, the design principles behind our ensemble of LLM agents are grounded in established research. Majority voting is a widely used strategy to improve LLM performance across various settings, including chain-of-thought reasoning \[1,3\], compound LLM systems \[2\], and text classification \[4\]. Motivated by these findings, we adopt a straightforward majority-voting scheme for our emotion-annotation pipeline. Our use of three medium-sized LLMs with comparable performance aligns with prior observations that ensemble methods are most effective when the individual models exhibit similar accuracy levels \[4\]. Third, unlike most benchmarks that only utilize limited evaluation metrics, we carefully design two complementary axes of evaluations for this task to make sure they capture all different types of errors (as summarized in Table 4).
>
> \[1\] Wang, X., Wei, J., Schuurmans, D., Le, Q., Chi, E., Narang, S., ... & Zhou, D. (2023). Self-consistency improves chain of thought reasoning in language models. ICLR.
> \[2\] Chen, L., Davis, J. Q., Hanin, B., Bailis, P., Stoica, I., Zaharia, M. A., & Zou, J. Y. (2024). Are more llm calls all you need? towards the scaling properties of compound ai systems. Advances in Neural Information Processing Systems (Neurips).
> \[3\] Choi, J., Yun, J., Jin, K., & Kim, Y. (2024). Multi-news+: Cost-efficient dataset cleansing via llm-based data annotation. EMNLP..
> \[4\] Trad, F., & Chehab, A. (2024, June). To ensemble or not: Assessing majority voting strategies for phishing detection with large language models. In International Conference on Intelligent Systems and Pattern Recognition (pp. 158-173). Cham: Springer Nature Switzerland.
>
> **4\. \[discussion\] Some claims about 'first of its kind' feel overstated given existing induced-emotion datasets (eg CMSV 2024).**
> Video2Reaction is indeed the first benchmark to map a video to induced emotion distribution in the wild. CMSV 2024 focuses on micro-videos and on a different task, predicting a single label of induced emotion given a video and a comment. Though our benchmark also focuses on induced emotion, our task is to predict a distribution of induced emotions given a video only.

---

> ### Author Response · Authors · 2025-11-21
> **(2/2) Initial Response**
>
> **5\. \[clarification\] How does the LLM ensemble handle cross-lingual comments?**
>
> All of our LLMs can handle multi-lingual text data. We will add this detail to the Appendix in the revised manuscript.
>
> **6\. \[clarification\] Can future updates dynamically re-label older clips to track temporal drift?**
>
> Yes, because each comment is timestamped, we can track temporal drift in the aggregated video-level reaction distribution based on the time-filtered comment-level data. Moreover, since our annotation pipeline is fully automated, these updates can be performed efficiently and at scale.
>
> **7\. \[methodology\] Were annotators or models ever confused by sarcasm or meme responses?**
>
> Yes, in our qualitative check in the initial phase, both annotators and LLM agents can be confused by sarcasm. Therefore, we added the first stage of rephrasing comments, which is not only crucial to filtering out irrelevant comments but also plays an important role in extracting an implicit stated reaction from sarcastic comments.
>
> **8\. \[discussion\] Could foundation models be fine-tuned efficiently using this dataset?**
>
> This is one of the future directions we would like to pursue with this dataset. While much of the current progress in video foundation models focuses on traditional video understanding tasks, we believe that equipping these models with the ability to forecast audience reactions is both powerful and feasible. Our experiments (Table 5 and 6\) already show that different types of label distribution learning (LDL) algorithms, when fine-tuned on our dataset, can improve meaningfully, suggesting that foundation models could similarly benefit from incorporating audience reaction prediction as an additional capability.

---

### Official Review · Reviewer_tQDy · 2025-11-01

**Soundness:** 2
**Presentation:** 3
**Contribution:** 2
**Rating:** 2
**Confidence:** 4

**Summary:**

The authors collect Video2Reaction, a dataset of 10K short movie segments from YouTube.  These segments are paired with a distribution over the categorical emotions that they elicit, derived from LLM-based labeling of their co-occurring comments.  They propose using Video2Reaction as a benchmark whose aim is predicting the distribution of audience reactions to video content.  They evaluate several different label distribution learning methods on the same set of multimodal features and a few video-language models on their benchmark using a selection of metrics they curate for the task.

**Strengths:**

The writing in this paper is very clear – it’s easy to follow what the authors did.

The author’s chosen problem, modeling the distribution of user emotion reactions, is an interesting one that is relatively understudied in the literature.

**Weaknesses:**

### Synthetic labels as a test set

A significant concern with the proposed benchmark is that the audience emotion distribution is synthetically derived from user comments.  While this would be fine for a training dataset, typically, for a test set, we want to ensure we have gold labels.  The authors perform a manual analysis of the their multi-LLM-predicted labels and find that they agree with 86% of them – what does a disparity of 86% at the label level imply for the noise in the derived audience emotion distributions?  Can the authors quantify that somehow?  What does it mean for the ceiling performance we should expect from models that we evaluate?

Additionally, it would be nice to see a confusion matrix for the emotion categories that were manually evaluated as some labels (like anger / embarrassment) were wrong 100% of the time.  That could very well be due to the small number of examples of anger in the manual evaluation set but that is not convincing on its own – would it be possible for the authors to manually annotate comments that reflect a balanced set of predicted emotions?

Another convincing result would be seeing that fine-tuning a video model on the training data in Video2Reaction results in a model that adapts well to a different audience emotion dataset with higher quality labels (for example, the work of Xu et al).  Then, we might have greater confidence in Video2Reaction’s derived labels.

### Weak baselines

While I am sympathetic to the compute demands of video modeling, a new video understanding benchmark demands evaluation against stronger baselines than those presented in this paper.  The multimodal features chosen are all quite dated / naïve (from 2020 – 2021) and the VLMs are fairly small.  The authors describe being unable to do model fine-tuning on their dataset – could they evaluate proprietary foundation models like Gemini to help us better understand what current ceiling performance is?  As is, it is difficult for researchers/practitioners who are versed in the strengths / weaknesses of the state-of-the-art models to make sense of this benchmark and its results.

### Lack of qualitative analysis

Typically, a new benchmark should be published with some qualitative analysis of model performance for others looking to pursue the task.  Which cases do models do well on?   Where do they struggle?  What makes this benchmark in particular hard?  This paper would benefit from this kind of analysis.

### Related work

The related work discusses many different methods for label distribution modeling but as the paper is not actually presenting a new method for label distribution modeling, this strikes me as unnecessary.  Instead, it would be nice to see the inclusion of more video datasets that leverage user comments (for comment generation or learning affective features) as these seem more related to the proposed task (modeling user responses).  Please see the list below:
- video comment generation: LiveBot (Ma et al, AAAI 2019), VideoIC (Wang et al, ACM 2020), HOTVCOM (Chen et al, ACL 2024), Personalized Video Comment Generation (Lin et al, EMNLP 2024)
- affective features learned from co-occurring video-comments: Enhancing Multimodal Affective Analysis with Learned Live Comment Features (Deng et al, AAAI 2025)

### Nitpicks
- when doing inline citations, please use \citet (for example in lines 316, 321)
- “foundation” typically refers to large-scale (and often proprietary) models like DeepSeek, GPT5, Claude, Gemini, etc; in this case, the authors are evaluating relatively small 7B parameter models

**Questions:**

The authors frame their derived emotion distributions as reflecting audience reactions to a particular video and they caveat (appropriately) in their limitations that this may not reflect global audience reactions but can they motivate the degree to which this reflects local audience reactions as well?  I would think there is a strong selection bias in who looks up / watches / comments on a particular video even with the same geographic area.  I don’t think this is a dealbreaker for the work but I do think it requires addressing.

The authors describing splitting videos into clips.  How are the comments for the video then associated with the segmented scenes?  Are all comments from the original video mapped to its parts?

What is the input to the VLMs you evaluate?  Is it just frames?  Do they get transcript information too?

To what degree does modeling reactions require prior movie context?  Viewers have likely seen all of the prior scenes/clips when re-watching a particular one -- their reactions likely reflect that prior knowledge.  Can the authors test this somehow?  Perhaps, if adding Gemini experiments, see if the model predicts the emotion distribution for scene t more accurately if it’s also seeing scenes 1 through t – 1.

---

> ### Author Response · Authors · 2025-11-21
> **(1/2) Initial Response**
>
> **1\. \[methodology\] A significant concern with the proposed benchmark is that the audience emotion distribution is synthetically derived from user comments. While this would be fine for a training dataset, typically, for a test set, we want to ensure we have gold labels.**
>
> This invites a discussion about how best to represent audience reaction, i.e. what should be the gold labels here. In our opinion, there are two main approaches to model audience reaction distribution, both valuable and complementary to each other: (1) recruit a small set of participants to watch the videos and provide reaction annotations (as done in LIBRIS-ACCEDE \[1\]) and (2) utilize social media’s self-reported reaction (ours and CMSV \[2\]). We opt for the second approach for two main reasons: (1) it enables us to capture reactions at a significantly larger scale and from a more diverse, global population, which is critical for modeling real-world variability in emotional response; and (2) social-media-based reactions reflect naturalistic viewing conditions, providing a more ecologically valid signal of how audiences engage with content outside of controlled lab settings.
>
> We do not claim that social-media-derived reactions should be the only metric for audience reaction modeling, but we argue that they are undeniably one of the key metrics and offer complementary insights to recruited viewers’ interviews and surveys. While the first approach is also valuable, they also introduce their own confounding factors (e.g., small sample sizes, demographic homogeneity, study-participant effects). Our dataset aims to complement existing efforts by providing another representation of audience reaction that would be difficult to obtain through small-scale annotation alone.
>
> \[1\] Yoann Baveye, Emmanuel Dellandrea, Christel Chamaret, and Liming Chen. Liris-accede: A video database for affective content analysis. IEEE Transactions on Affective Computing, 6(1):43–55, 2015\.
> \[2\] Cong Xu, Lu Liu, Liang Jin, Guoguang Du, Zhenhua Guo, Yaqian Zhao, Xuanjing Huang, Rengang Li, et al. Infer induced sentiment of comment response to video: A new task, dataset and baseline. Advances in Neural Information Processing Systems, 37:103737–103750, 2024\.
>
> **2\. \[methodology\] There is a strong selection bias in who looks up / watches / comments on a particular video even with the same geographic area.**
> We acknowledge that like any other social media-based data collection studies, our benchmark is subject to selection bias. However, we believe that modeling the reactions based on the population of commenters is the closest attempt to modeling the reactions of the population. Additionally, from the pragmatic standpoint of content creators, their primary interest is in understanding the reactions of users who actively engage with their content, as this group provides the most informative and actionable feedback.
>
> **3\. \[experiments\]  fine-tuning a video model on the training data in Video2Reaction results in a model that adapts well to a different audience emotion dataset with higher quality labels (for example, the work of Xu et al)**
> We thank the reviewer for this good suggestion. Exploring the transfer learning capability of models trained on Video2Reaction is a good future research direction. We are contacting the authors of that dataset to see if we can access raw videos to set up the evaluation experiment. The current release on their GitHub repository only includes I3D visual features (which are quite outdated; newer VideoMAE features are reported in the paper but not released yet) and does not include audio track or text description features that our baseline models are trained with. They also use a different taxonomy of emotions that requires nontrivial mapping to our 21-category emotion taxonomy. We will report back if we can complete this experiment in the time allowed.
>
> However, we would like to view this experiment more as an extension to showcase the application of the dataset rather than a necessary evaluation of our dataset’s quality. We would like to emphasize the main contribution of our paper to establish the first and comprehensive benchmark for modeling audience reactions in the wild with a fine-grained reaction taxonomy.
>
> **4\. \[experiments\] could they evaluate proprietary foundation models like Gemini to help us better understand what current ceiling performance is?**
> Thank you for the suggestion. We are in the process of getting the results for Gemini. There is a rate limit of 8 hours of video a day so we will update the results in a later comment after the experiment finishes.

---

> > ### Author Response · Authors · 2025-11-21
> > **(2/2) Initial Response**
> >
> > **5.\[experiments\] Which cases do models do well on? Where do they struggle? What makes this benchmark in particular hard?**
> > This benchmark is particularly hard because predicting a distribution over 21 discrete categories is much more difficult than predicting one single dominant label. In Appendix D.1 Figure 4, we show some qualitative visualization of how the predicted distribution of two leading algorithms differ from the ground truth distribution across different levels of entropy. We also demonstrate the current STOA algorithms struggle with long-tail problems (Appendix D.2 Table 14). If there are specific additional analyses you would like to see, please let us know.
> >
> > **6\. \[clarification\] What is the input to the VLMs you evaluate? Is it just frames? Do they get transcript information too?**
> > Inputs to models are embedded key frames of the video (main modality), and auxiliary modalities, including the embeddings of the extracted audio track, and the text embeddings of the short overall description of the movie clip. Transcript information can be further extracted from the audio track, but since the audio track has already been provided to the model, it is relatively redundant.
> >
> > **7\. \[clarification\] How are the comments for the video then associated with the segmented scenes?**
> > The splitting/segmentation is done and uploaded to YouTube by the Movieclips YouTube channel; the comments are then collected from the comment area under each uploaded movie clip video's YouTube page, so there is a natural mapping between the comments and the movie clip.
> >
> > **8\. \[related work\] The related work discusses many different methods for label distribution modeling but as the paper is not actually presenting a new method for label distribution modeling, this strikes me as unnecessary.**
> >
> > We identify the learning target in our dataset as a label distribution that reflects more complex and mixed emotional feedback from the YouTube users. Therefore, it is natural and necessary to discuss existing label distribution learning literature to serve as preliminary for our evaluation metrics and baseline methods. We think that the related work section on label distribution learning together with the baseline LDL methods in our benchmark will encourage future researchers to advance in this under-studied field.
> >
> > **9\. \[related work\] it would be nice to see the inclusion of more video datasets that leverage user comments (for comment generation or learning affective features) as these seem more erelated to the proposed task (modeling user responses).**
> > We thank the reviewer for suggesting a potential new application for our dataset. Our benchmark focuses on discovering audience reaction distribution, not generating synthetic comments so we do not include these works in our literature review. However, it is a good potential downstream application of our dataset. We will include these works in our related work section and investigate the use of our dataset for the task of comment generation in the future.
> >
> > **10\. \[discussion\] To what degree does modeling reactions require prior movie context?**
> > We explored the variation in the audience reaction distribution between clips of the same movie and observed a substantial variation (Section 3.4, Line 245-257). This suggests that reactions are often driven by the local content of the clip itself rather than solely by broader movie context. Further analyses to establish the causal relationship between prior movie context and reactions would be interesting in future work.

---

> > ### Comment · Reviewer_tQDy · 2025-11-21
> >
> > Thanks for your engagement with my review.  I will come back and reply to your remaining responses soon but wanted to call this out as I think there's some misunderstanding that I'd like you to have the time to address.
> >
> > Re 1: my issue is not so much that you're using emotions derived from the comment distribution as the label in this task (that seems reasonable / there's not really other ways to get at that data); I think if you discuss the limitations of that explicitly in the paper, I am satisfied (as I raise and you address in 2).  My concern is the mechanism for deriving those emotions -- the model based labeling of the emotionality of each comment.  Could you address the rest of my comment whose focus is that (about the error rate, the implied ceiling, confusion matrix, 0 performing labels like anger).  If these synthetically derived labels are going to be used as a test target in evaluation, it's important that we have high confidence in their quality.
> >
> > I see another reviewer has reasonable concerns about the structure of the validation task (asking for verification instead of labeling); I am not entirely convinced by your response -- I think if you asked humans to label and the model had IAA similar to humans that would be satisfactory, yes?  Even if the task is one with low IAA?

---

> > > ### Author Response · Authors · 2025-11-21
> > >
> > > Thank you for a quick response and the clarification. I thought in the original comment, you referred to using comments as a construct for audience reaction. I will address your question about the impact of error rate on the ceiling performance of our evaluation metrics, including both distributional metrics like KL Divergence and classification metrics like Top-k F1 score in a later comment. Quantifying the ceiling performance for classification metrics is straightforward but a bit more tricky for distribution metrics.
> > >
> > > Regarding your second question, do you mind clarifying this part:
> > > > if you asked humans to label and the model had IAA similar to humans that would be satisfactory, yes?
> > >
> > > What do you mean by the IAA of the model? Are you referring to the IAA of our ensemble of LLM agents or something else? I would like to make sure I understand your question correctly before addressing it.

---

> ### Comment · Reviewer_tQDy · 2025-11-22
>
> Of course, thanks again for your engagement.
>
> Re: the IAA, I mean, if the model's agreement with humans is similar to human agreement among themselves in the open label setup, that would be stronger evidence for the quality of the dataset than the verification protocol you employ as it's less likely to induce a bias in your human judgments.
>
> Alternatively, if it really is a task that humans cannot do well (necessitating your verification setup), perhaps you can assign random emotion labels to comments and ask humans to verify them so we have some sense of the bias of the underlying protocol.

---

> > ### Author Response · Authors · 2025-11-24
> >
> > Thank you for the clarification and suggestion. We are doing the human verification test with LLM vs. random label and we will report here once we finish and get the results.
> >
> > Now I would like to address the ceiling performance question and the Gemini experimental result.
> >
> > **11. [discussion] The authors perform a manual analysis of their multi-LLM-predicted labels and find that they agree with 86% of them – what does a disparity of 86% at the label level imply for the noise in the derived audience emotion distributions? Can the authors quantify that somehow? What does it mean for the ceiling performance we should expect from models that we evaluate?**
> >
> > Regarding the distribution metrics:
> > The ceiling performance for Chebyshev (defined as the maximum per-class prediction error) is $0.14$. The current STOA performance is $0.2306$ so there is still a good gap between the ceiling performance and STOA.
> >
> > The ceiling performance for KL divergence is $0.1508$ and the current STOA performance is $0.5976$. Let $p$ be the true video-level distribution and $q$ be our annotated distribution. Assume that a comment is annotated with an error rate $\epsilon = 0.14$ and that the wrong label is uniformly sampled from the rest of the classes, the annotated distribution $q$ is upper-bounded by $-\log(1-\epsilon) = 0.1508$.
> >
> > $$KL(p|q) = \sum_ip_i\log\frac{p_i}{q_i} \leq \sum_ip_i\log\frac{p_i}{(1 - \epsilon)p_i} = \sum_ip_i(- \log (1 - \epsilon) = - log(1 - \epsilon) = 0.1508$$
> >
> > Regarding the dominant reaction distribution metrics:
> > The ceiling performance for top-1 F1 score is $0.86$ and the current STOA is $0.5283$ so there is still more room for improvement.
> >
> > **4. [experiments] Experimental results on propriety foundation models like Gemini**
> > We provided both unscaled and temperature-scaled evaluation results for Gemini 2.5 here. The performance is better than other VLMs but still underperforms compared to specialized algorithms SA-BFGS.
> >
> > ## Gemini: Distribution Metrics
> >
> > | Method | Cheb | Cla | KL | Cad | Inter | Cos |
> > | --- | --- | --- | --- | --- | --- | --- |
> > | gemini 2.5 flash (unscaled) | 0.3483 | 3.4008 | 4.5526 | 3.2840 | 0.3659 | 0.5095 |
> > | gemini 2.5 flash (temperature scaled) | 0.3425 | 3.4477 | 4.8102 | 3.4316 | 0.3787 | 0.5029 |
> >
> > ## Gemini: Dominant Reaction Metrics
> >
> > | Method | MRR | F1 Top 1 (weighted) | F1 Top 2 (weighted) | F1 Top 3 (weighted) | TPE |
> > | --- | --- | --- | --- | --- | --- |
> > | gemini 2.5 flash (unscaled) | 0.4378 | 0.2735 | 0.3332 | 0.3794 | 0.3282 |
> > | gemini 2.5 flash (temperature scaled) | 0.4378 | 0.2735 | 0.3332 | 0.3794 | 0.3026 |

---

> > > ### Author Response · Authors · 2025-11-26
> > > **results for the randomly assigned labels**
> > >
> > > Thank you again for the suggestion. We just completed this test on a random sample of 100 comments. We presented the reviewer with two options, one is LLM-annotated label and one is randomly assigned labels, and the reviewer is not aware which one is which. The result below shows that (1) human verification approach can distinguish between correct and incorrect labels and not subject to confirmation bias and (2) LLM-annotated labels are preferred in most cases.
> > >
> > > | Category              | Value  |
> > > |-----------------------|--------|
> > > | Total Responses       | 100    |
> > > | LLM Preference %      | 87.00% |
> > > | Random Preference %   | 2.00%  |
> > > | Neither Preferred %   | 11.00% |

---

### Meta-Review · Area_Chair_FogA · 2026-01-09

**Summary:**

This paper introduces Video2Reaction, a large-scale multimodal dataset aimed at modeling audience emotional responses to video content, rather than emotions expressed by characters or intended by creators. The paper received four reviews (one review has been flagged as possibly LLM-generated by the original AC), with three reviewers expressing substantive concerns that were not fully resolved during rebuttal, and no reviewer intended to increase their score according to the reviewer-author discussion. The primary unresolved issues center on the potential bias of the verification protocol and the reliability of fine-grained emotion annotations, both of which directly affect the validity of the dataset. While the motivation is interesting, these concerns remain significant and limit confidence in the proposed benchmark in its current form.

**Reviewer Concerns:**

The main concerns are the potential bias of the verification protocol and the reliability of fine-grained emotion annotations. According to the discussion, the reviewers are not fully convinced by the rebuttal.

**Reviewer Scores:**

According to the discussion, no reviewer intended to increase their score.

---

### Decision · Program_Chairs · 2026-01-26

Reject